# Beyond Benchmarks: Toward Causally Faithful Evaluation of Large Language Models

Zhengshuyuan Tian [* 1 2]  Wanling Gao [* ✉ 1 2 3]  Chuanxin Lan [1]  Chenxi Wang [1 2]  Lei Wang [1 2 3]  Guoxin Kang [1 3]  Zhengxin Yang [1 3]  Yunyou Huang [4]  Xuehai Hong [✉ 1 2]  Jianfeng Zhan [1 3]

## Abstract

Current LLM evaluations often conflate benchmark performance with intrinsic model capability. This is misleading, as observed outcomes arise from the entire evaluation system, including datasets, prompting methods, decoding parameters, and the software–hardware stack, rather than the model alone. When this system is underspecified, attribution becomes unreliable; in practice, evaluation choices alone can induce accuracy swings of up to 70%. This challenge is compounded by the open-ended nature of LLM evaluation, where questions span languages, domains, and usage styles, forming variable and implicitly shifting datasets. Consequently, strong performance on static benchmarks may reflect surface alignment or dataset-induced effects rather than robust capability. Prior studies often focus on individual components or manually-curated small-scale dataset variants, overlooking interactions and dataset-related confounding. To address these limitations, we propose LLM evaluatology, a principled framework that grounds LLM evaluation in a causally motivated system design. It combines structured causal modeling as an intervention-oriented lens with factorial decomposition under design of experiments, quantifying main and interaction effects while using instance-level interventions to probe dataset-induced effects. By jointly modeling evaluation components and structured question variations, LLM evaluatology enables more interpretable, reproducible, and carefully attributed assessment of model capability. Our framework is publicly available at `GitHub`.

---

[*]Equal contribution [1]Institute of Computing Technology, Chinese Academy of Sciences [2]University of Chinese Academy of Sciences [3]International Open Benchmark Council [4]Guangxi Normal University. Correspondence to: Wanling Gao <gaowanling@ict.ac.cn>, Xuehai Hong <hxh@ict.ac.cn>.

*Proceedings of the 43ʳᵈ International Conference on Machine Learning*, Seoul, South Korea. PMLR 306, 2026. Copyright 2026 by the author(s).

## 1. Introduction

Current LLM evaluation practices are fragmented and ad-hoc, spanning standardized test–style benchmarks (Hendrycks et al., 2021a; Huang et al., 2023; Rein et al., 2024; Suzgun et al., 2023; AIME, 2025), human preference–based benchmarks (Chiang et al., 2024; OpenCompass, 2025; Xu et al., 2023), and dynamic or continuously refreshed benchmarks (Jain et al., 2025; Jimenez et al., 2024; White et al., 2025; Zhu et al., 2024; Li et al., 2025). Yet all largely treat the model in isolation, neglecting that measured performance arises from the entire evaluation system, including dataset, prompts, decoding, and even the software–hardware stack. In reality, LLM evaluation is inherently a high-dimensional problem, as these interacting components jointly shape outcomes and complicate attribution. As recent studies show, results can vary sharply with dataset artifacts (Nguyen et al., 2025; Liu et al., 2026), prompting (He et al., 2024; Biderman et al., 2024), decoding (Shi et al., 2024; Song et al., 2025), or annotator biases (Das et al., 2024). But such analyses remain piecemeal, each targeting a single component without quantifying their combined impact or enabling principled attribution. What is missing is a rigorous framework that disentangles intrinsic model capability from confounding influences.

Even under a fully specified evaluation system, LLMs differ fundamentally from traditional deterministic models. Their usage involves open-ended questions spanning languages, domains, and usage styles, ranging from instances similar to training data to those requiring analogical reasoning or entirely novel generalization. This diversity undermines the notion of "typical" questions, rendering evaluation on a small set of canonical examples insufficient. Moreover, because LLMs are trained on massive and heterogeneous corpora, evaluation datasets may partially overlap—directly or indirectly—with pretraining data, introducing dataset-induced confounding between model capability and observed performance. Thus, correctness on individual instances may reflect surface familiarity rather than genuine competence. Consequently, reliable evaluation must account for systematic question variation to disentangle intrinsic capability from surface-level correctness, while carefully controlling

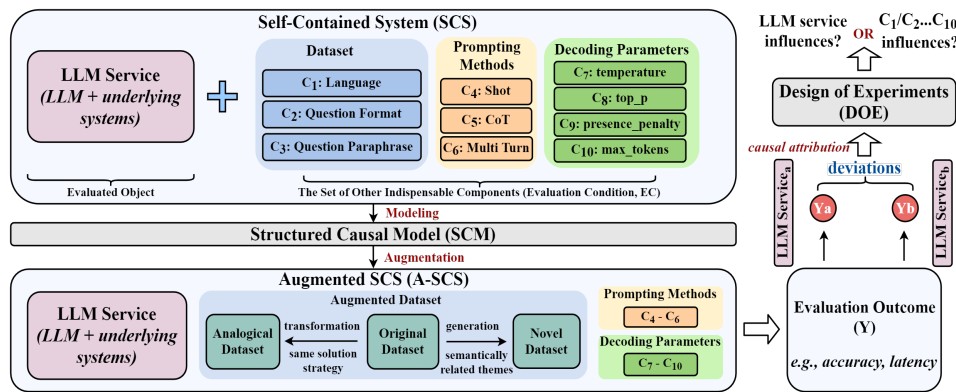

*Figure 1.* LLM Evaluatology: a principled framework that grounds LLM evaluation in a causally motivated system design. Here, the evaluated object is defined as the LLM service, comprising the LLM and its underlying systems. When evaluating a standalone LLM, the underlying systems are instead treated as part of the evaluation conditions (EC).

for confounding effects within the evaluation system.

This paper introduces LLM evaluatology (Figure 1), a systematic framework for LLM evaluation that integrates structured causal modeling (SCM) (Pearl, 2009) with factorial decomposition under design of experiments (DOE) (Montgomery, 2017; Telford, 2007) to support principled attribution. Building on Evaluatology (Zhan et al., 2025; 2024), we formalize a Self-Contained System (SCS) that specifies the evaluated object (e.g., a standalone LLM or an LLM service), the indispensable components shaping measured performance, and the evaluation conditions (EC)—i.e., the admissible configuration space induced by those components. Making this system explicit is the prerequisite for any interpretable attribution: without a specified evaluation system, benchmark scores confound model capability with unspecified procedural choices. Within the SCS, SCM formalizes evaluation assumptions, causal structure among components, and intervention targets, treating outcomes as jointly shaped by model capability and EC configurations. It serves as an intervention-oriented lens, not as a claim of complete causal identification from observational logs alone. A key consequence is that pretraining–evaluation overlap can render benchmark exposure non-innocuous, opening backdoor-style paths through dataset-related dependencies and motivating instance-level treatment of dataset effects. Augmented SCS (A-SCS) responds to this challenge by constructing seen, analogical, and novel evaluation variants that probe dataset-induced variation while limiting sensitivity to overlap and surface familiarity. DOE over the resulting controllable system then quantifies main and interaction effects across prompting, decoding, and dataset-related components. LLM evaluatology thus integrates system specification, causal reasoning, instance-level intervention, and factorial decomposition into a reproducible and automatable foundation for systematic evaluation analysis.

Our experiments show that LLM performance depends on the full evaluation system rather than the model alone: under SCS/A-SCS, the same model's accuracy ranges from 0 to 0.93 and can even reverse rankings, rendering single-configuration evaluations unreliable. By jointly modeling ten essential components, we find that a few components—most notably Question Format and CoT, followed by max_tokens, Shot, Multi Turn, and Language—and their interactions explain most performance variance, with strong model-specific sensitivities. Notably, even components with minor individual effects can combine to produce substantial variance. Systematic sampling over this configuration space allows LLM evaluatology to produce accuracy estimates that closely approximate true performance across all conditions.

## 2. Related Work

We group existing benchmarks into three types. Standardized test–style benchmarks evaluate model outputs against reference answers. For example: general knowledge and reasoning (MMLU (Hendrycks et al., 2021a), MMLU-Pro (Wang et al., 2024b), MMLU-Redux (Gema et al., 2025), C-Eval (Huang et al., 2023), CMMLU (Li et al., 2024), and GPQA (Rein et al., 2024)); reasoning and commonsense (BBH (Suzgun et al., 2023), HellaSwag (Zellers et al., 2019), and Winogrande (Sakaguchi et al., 2021)); mathematics (GSM8K (Cobbe et al., 2021), MATH (Hendrycks et al., 2021b), and AIME (AIME, 2025)); coding (HumanEval (Chen et al., 2021), MBPP (Austin et al., 2021), Aider-polyglot (Aider, 2025), and MultiPL-E (Cassano et al., 2023)); long-context understanding (L-Eval (An et al., 2024), LongBench (Bai et al., 2024), ∞Bench (Zhang et al., 2024a), and HELMET (Yen et al., 2025)); safety (SafetyBench (Zhang et al., 2024b) and Toxigen (Hartvigsen et al., 2022)); instruction-following (IFEval (Zhou et al., 2023) and multi-challenge (Deshpande et al., 2025)); and multimodal benchmarks (MMBench (Liu et al., 2024), MMMU

(Yue et al., 2024), and MathVista (Lu et al., 2024)).

Human preference–based benchmarks evaluate models in interactive settings, collecting user judgments instead of relying on fixed test sets. Chatbot Arena (Chiang et al., 2024) is the most prominent example, where pairwise votes are aggregated via Elo ratings. CompassArena (OpenCompass, 2025) applies similar designs in the Chinese context.

Dynamic or continuously refreshed benchmarks aim to avoid data contamination by relying on newly released or procedurally generated datasets. Examples include Live-CodeBench (Jain et al., 2025), SWE-bench (Jimenez et al., 2024), LiveBench (White et al., 2025), DyVal (Zhu et al., 2024), and Arena-Hard (Li et al., 2025).

Prior work has shown that one or a few components, such as prompts and formatting (Biderman et al., 2024; Sclar et al., 2024; Lunardi et al., 2025; Haase et al., 2026; Kunievsky & Evans, 2025), language (Singh et al., 2025), or decoding hyperparameters (Song et al., 2025), can substantially affect reported LLM performance, motivating standardized evaluation harnesses for reproducibility (Biderman et al., 2024; Reuel et al., 2024). Manual, small-scale manipulations, like introducing distractors or numerical modifications, offer preliminary evidence that models often rely on surface patterns rather than robust reasoning (Mirzadeh et al., 2025), yet fail to account for the rich variability and nuanced linguistic features of human-authored questions. While informative, these studies are fragmented: they assess limited components in isolation and do not quantify how interacting components jointly shape outcomes, nor provide a framework to disentangle model capability from evaluation artifacts.

## 3. Motivation

**The flaw of existing LLM evaluation methodology.** Existing LLM benchmarks define dataset formats and scoring rules, but leave crucial indispensable components uncontrolled, e.g., decoding parameters and prompting methods. We systematically reviewed major benchmarks and compiled a taxonomy of which components are explicitly defined and which are left open (Table 1). Strikingly, many widely used benchmarks, including AIME, specify only a subset of variables while leaving key components underspecified. To quantify the implications, we reconstructed the AIME evaluation space by enumerating plausible settings of uncontrolled components (e.g., CoT, temperature, top_p, presence_penalty, max_tokens), yielding 162 distinct evaluation conditions. Accuracy under these conditions varied by as much as 70% across settings, and the distributions often diverged substantially from the single numbers reported in technical documentation; on some models, the median relative change between our measured accuracy and the accuracy reported in the technical report reached as high

as 50% (Figure 2). Comparable inconsistencies are evident in MMLU (Appendix A.1) and other flagship benchmarks, suggesting that the problem is not benchmark-specific but structural across current LLM evaluation methodologies.

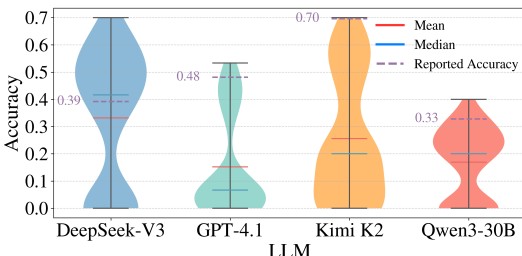

*Figure 2.* Accuracy Deviations on AIME When Evaluating with Identical Dataset across 162 Combinations of Component Settings.

Our experiments reveal that interactions between components can substantially amplify effects. For instance, on GPT-3.5 (Appendix A.2), neither Shot ($p = 0.58$) nor Multi Turn ($p = 0.87$) alone significantly affects performance, yet their interaction is highly significant ($p = 0.01$), demonstrating that the combined influence of otherwise negligible components can meaningfully impact outcomes. This underscores the necessity of analyzing component interactions within the full evaluation system, which single-component studies (Biderman et al., 2024; Singh et al., 2025; Song et al., 2025) fail to capture.

**The challenges of using Evaluatology for LLM evaluation.** Extending Evaluatology (Zhan et al., 2025; 2024; Wang et al., 2024a) to LLMs is qualitatively challenging: usage is open-ended, so the effective "unit of evaluation" drifts across users, languages, and time. In our experiments, when evaluating a question on nine LLMs including DeepSeek, Doubao, Kimi, Mistral, Qwen and GPT series, etc., five were able to solve this original (seen) question correctly. However, after performing analogical transformations through inserting distractor, none of them produced correct solutions. This striking contrast illustrates why A-SCS is essential: performance on a single question can be misleading, as models may succeed on problems they have effectively memorized yet fail when the same reasoning must be applied under slightly altered conditions.

## 4. LLM Evaluatology

LLM evaluatology consists of four essential steps: (1) defining SCS, (2) specifying SCM-oriented causal structure, (3) constructing A-SCS, and (4) evaluating SCS/A-SCS and attributing evaluation outcomes.

### 4.1. Defining Self-Contained System (SCS)

We define the Self-Contained System (SCS) for LLM evaluation as the smallest independently runnable system that

*Table 1.* Evaluation Settings across Benchmarks. (Lang.=Language; Fmt.=Question Format; Para.=Question Paraphrase; CoT=chain-of-thought; MT.=Multi Turn; Temp.=temperature; PP=presence_penalty; MaxTok.=max_tokens; Eng. = English; Mul. = Multiple language; ori=original; y/n/x=yes/no/not reported).

| Benchmark | Dataset | | | Prompting | | | Decoding | | | |
| --- | --- | --- | --- | --- | --- | --- | --- | --- | --- | --- |
| | Lang. | Fmt. | Para. | Shot | CoT | MT. | Temp. | top_p | PP | MaxTok. |
| MMLU | Mul. | ori | n | 0/3/5 | y/n | n | 0.0/0.3/0.6/0.7 | 0.8/0.95 | 0.5/1.5 | 8192/32768 |
| AIME | Eng. | ori | n | x | y/n | n | 0.0/0.6/0.7 | 0.8/0.95 | 0.5/1.5 | 8192/32768/38912 |
| GPQA | Eng. | ori | n | 0/5 | y/n | n | 0.0/0.6/0.7 | 0.8/0.95 | 0.5/1.5 | 8192/32768 |
| MATH | Eng. | ori | n | 0/4/8 | y/n | n | 0.0/0.6/0.7 | 0.8/0.95 | 0.5/1.5 | 8192/32768 |
| SWE-bench | Eng. | ori | n | 0 | y/n | n | 0.0/0.6 | 0.95 | x | 8192/16384/32769 |
| IFEval | Eng. | ori | n | 0 | y/n | n | 0.0/0.6/0.7 | 0.8/0.95 | 0.5/1.5 | 8192/32768 |
| Arena-Hard | Eng. | ori | n | 0 | y/n | n | 0.0/0.6/0.7 | 0.8/0.95 | 0.5/1.5 | 8192/32768 |
| HumanEval | Eng. | ori | n | 0 | y/n | n | 0.6 | 0.95 | x | 8192/32768 |

includes the evaluated object and all indispensable components that materially affect the evaluation outcome. The evaluated object $O$ is not limited to a bare LLM; it can also encompass the broader deployed LLM service that fuses the model with its supporting software and hardware stack. For example, when evaluating through an API, the LLM and its underlying systems should be treated as an inseparable whole, whereas for locally deployed open-source models, the surrounding system environment may either be incorporated into $O$ or explicitly modeled as part of the other indispensable components.

After defining the evaluated object, the second step is to identify the indispensable components that shape evaluation outcomes and to establish their value ranges, collectively denoted as evaluation conditions (EC). We organize EC into three layers and together yield 10 key components ($C_1$–$C_{10}$). *Dataset* captures data-related variations, including Language, Question Format, and Question Paraphrase ($C_1$–$C_3$). Note that Question Paraphrase is introduced as a key component to mitigate hallucination and data contamination, referring to reformulating questions without altering their semantics or correct answers. *Prompting methods* account for interaction styles, namely Shot, CoT (chain-of-thought), and Multi Turn ($C_4$–$C_6$). *Decoding parameters* represent inference controls, including temperature, top_p, presence_penalty, and max_tokens ($C_7$–$C_{10}$). Each component is instantiated with representative values to balance coverage of real-world variability against configuration space tractability, as summarized in Table 2. Note that some seemingly "extreme" values like max_tokens=10 and temperature=2.0 are included to capture diverse user requirements, e.g., compact output formats and creative generation; ablation studies confirm that the seemingly "extreme" evaluation results are not caused by these "extreme" settings (details in Appendix A.3). The SCS is then specified as $EC \times O$, ensuring that performance measurements are attributed correctly while systematically controlling for confounding factors introduced by indispensable components.

*Table 2.* Evaluation Conditions: Indispensable Components and Value Ranges.

| Component | Value Range |
| --- | --- |
| Language | Chinese, English, Japanese, Arabic, French, Russian |
| Question Format | Multiple-choice, Fill-in-the-blank |
| Question Paraphrase | Yes, No |
| Shot | Yes, No |
| CoT | Yes, No |
| Multi Turn | Yes, No |
| temperature | 0.0, 1.0, 2.0 |
| top_p | 0.2, 0.6, 1.0 |
| presence_penalty | −0.5, 0.5, 1.5 |
| max_tokens | 10, 100, 4000 |

### 4.2. Specifying SCM-Oriented Causal Structure

Given SCS, we specify a *causal directed acyclic graph (DAG)* over the aggregated components implied by $EC$ (Figure 3). Concretely, we group components related to dataset ($C_1$–$C_3$), prompting ($C_4$–$C_6$), and decoding ($C_7$–$C_{10}$) into $D$, $P$, and $\Theta$, respectively. Let $O_M$ denote *model capability*: the portion of benchmark performance attributable to the evaluated LLM, rather than to EC choices. Let $Y$ denote the *evaluation outcome*. The causal DAG we use is

$$D \to Y, \quad P \to Y, \quad \Theta \to Y, \quad O_M \to Y, \quad D \to O_M,$$

where each arrow denotes a direct causal effect: $D$, $P$, and $\Theta$ may shift $Y$, and $D$ may additionally shift $O_M$ when benchmark exposure is non-innocuous (e.g., via pretraining–evaluation overlap). Consequently, $O_M \leftarrow D \to Y$ forms a backdoor path: when dataset-side variation confounds both, attributing $Y$ solely to $O_M$ is generally misleading.

### 4.3. Constructing Augmented SCS (A-SCS)

A-SCS preserves the original questions (*seen*) within the dataset and augments them through **analogical transformation** and **novel generation**. In our implementation, these two augmentations are realized by five systematically defined, script-driven pipelines in total. For each pipeline, we fix general prompts and scaffolding, and then run the

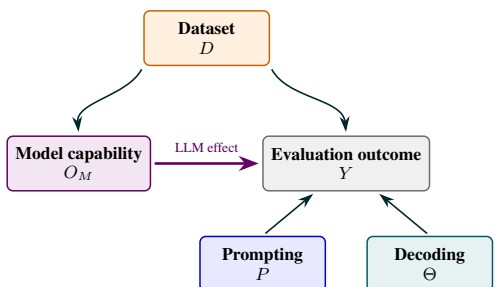

*Figure 3.* SCM-Oriented Causal Structure over SCS.

entire process automatically through LLM API calls (e.g., GPT-5), lightweight verification scripts, and other auxiliary tooling. This setup scales to large datasets and produces diverse variants, without any per-question manual rewriting or hand-crafting (Figure 4).

### 4.3.1. ANALOGICAL TRANSFORMATION PIPELINES

**Distractor Insertion** augments an original question by adding redundant sentences at random positions. We systematically divide redundant information into three categories: (i) context-irrelevant redundancy; (ii) context-relevant explanatory redundancy that explains concepts already appearing in the question; and (iii) context-relevant misleading redundancy that is logically related to the question but deliberately nudges the solver toward an incorrect strategy. By providing the LLM with transformation examples, correct answers, and (optionally) solution steps, all three types of redundancy are generated via similar structured prompts and automatically inserted at random positions.

**Numeric Substitution** augments a question by systematically perturbing its key numerical parameters. Using the correct answer, solutions, and related formulas, the LLM is prompted to generate a Python solver that explicitly parameterizes the key numbers in the question. The system automatically wraps the generated `solve` function into a runnable script, executes it via `subprocess`, and feeds any errors or mismatches back to the LLM for refinement until verification. It then samples new parameters within a reasonable range and reuses the solver to generate a family of numeric variants without manual intervention.

**Conditional Recomposition** augments a question by constructing "inverse" variants where the original answer is treated as a given condition and some of the original conditions become the new target quantities. We first prompt the LLM to identify which condition and target quantity can be interchanged and generate a new question. Optionally, following a procedure similar to numeric substitutions, a Python solver that explicitly parameterizes the key numerical quantities can be generated and iteratively refined until it passes verification. Once a verified solver is obtained, the new input conditions may be perturbed within a reason-

able range to automatically produce multiple "conditionally recomposed" numeric variants of the original question.

### 4.3.2. NOVEL GENERATION PIPELINES

**Recent-source Adaptation** augments a question by aligning it with thematically similar questions drawn from recent real-world exams. Given an original question, we first use an LLM to extract its core knowledge points. We then query an open online exam-question banks, indexed by year, region, subject, and knowledge point, to retrieve recent (e.g., within three months) exam questions that match these knowledge points. The retrieved questions are subsequently paraphrased via LLM, and can optionally be further transformed using the three analogical pipelines described above. In this way, we obtain recent-source adapted questions that remain aligned with the original question at the knowledge-point level while being entirely new instances.

**Conceptual Synthesis** augments a question by generating conceptual questions that target the underlying concepts. Based on authoritative textbooks in PDF form, we build a structured knowledge base in which each concept is associated with its definitions, theorems, phenomena, and canonical examples extracted from the textbooks. For a given question, we use an LLM to identify its primary knowledge points and retrieve the corresponding concept entries from the knowledge base. We then prompt the LLM to synthesize new conceptual questions grounded in these entries, yielding questions that probe conceptual understanding underlying the origin question.

Detailed procedures, examples, and algorithmic pseudocode on A-SCS are provided in Appendix A.4.

### 4.4. Evaluating and Attributing Evaluation Outcomes

SCS samples the full configuration space defined by 10 components. A-SCS extends this space by applying five augmentation pipelines, comprising seven augmentation mechanisms in total, to each question within a dataset, filtering out failed transformations (e.g., numeric substitutions without a stable solver).

Given the exponentially large configuration space, exhaustive evaluation is generally infeasible. Evaluation under SCS/A-SCS therefore balances accuracy and cost by systematically sampling configurations from a pre-generated admissible list. Specifically, we generate the full configuration list once, shuffle it with a fixed random seed, and share the resulting order across all models. For each model, an evaluation budget of size $N$ corresponds to evaluating the first $N$ configurations in the shared order, without reshuffling. The budget $N$ is determined by two complementary criteria. First, *before* the main evaluation run, we use variance estimated from pre-specified pilot samples to

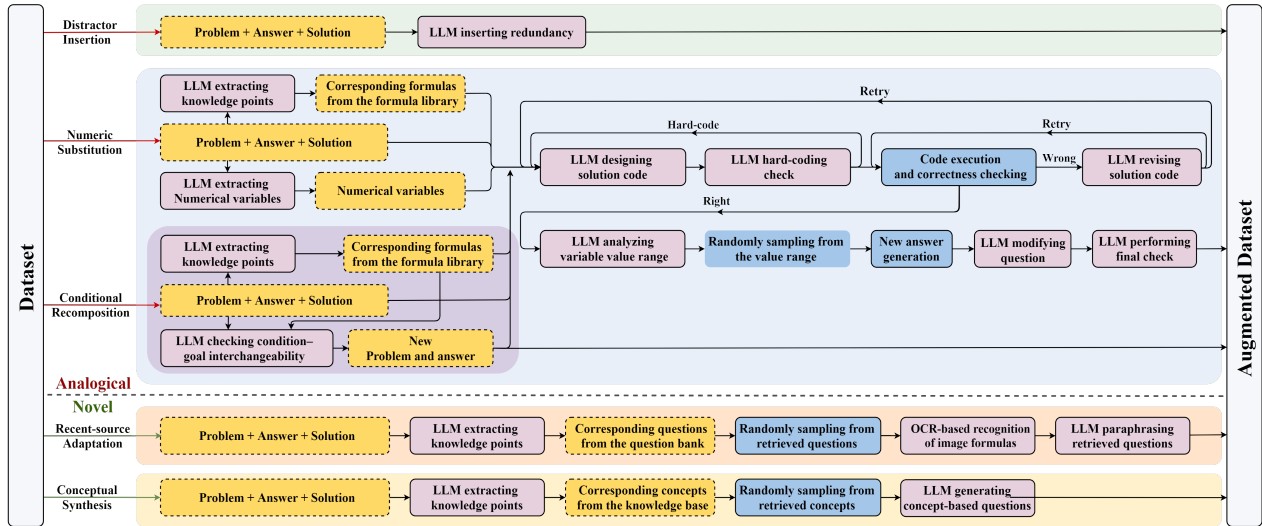

*Figure 4.* Constructing Augmented SCS (A-SCS): Analogical Transformation and Novel Generation Pipelines.

apply a classical normal-approximation calculation, yielding a sample-size floor $N_{NA}$ required to meet a target error tolerance at a given confidence level. Second, *during* evaluation, configurations are processed sequentially in fixed-size batches (e.g., 10) along the same order; after each batch, we recompute the running mean accuracy and an ordinary 95% confidence interval over all configurations evaluated so far, and stop when (i) the absolute changes in the running mean over the last three updates are all below 0.002, and (ii) the 95% interval width is below 0.06, yielding $N_{conv}$. The final budget is $N = \max(N_{conv}, N_{NA})$ and the evaluation continues along the same order until $N$ configurations are reached. The convergence criterion is used only as a practical tool for monitoring stability and is not anytime-valid, hence it does not by itself provide formal sequential coverage guarantees. Nevertheless, the overall procedure is statistically well-motivated and the resulting sample sizes are empirically stable, statistically sound, and cost-effective.

The attribution procedure proceeds in three steps:

(1) Obtaining overall effect through EC sampling. The evaluation system is sampled across diverse configurations, producing performance outcomes for the evaluated model under a broad spectrum of settings. The mean performance across these samples, together with 95% and 99% confidence intervals, provides a stable summary metric that balances comprehensiveness with practical efficiency.

(2) Attributing individual component effects. To attribute effects to a specific component, all other components are held constant while varying only the one of interest (equivalent evaluation conditions). Differences in measured performance can then be directly attributed to that component, mitigating confounding from other components.

(3) Quantifying variance contributions via ANOVA under

DOE. Analysis of variance (ANOVA) is applied across the sampled configurations to estimate the proportion of performance variance explained by each component. This statistical approach complements controlled experiments, enabling systematic, quantitative attribution of component effects on overall model performance.

## 5. Evaluation

### 5.1. Experimental Setup

**Target Models.** In this section, we evaluate our methodology on mainstream publicly accessible LLMs: DeepSeek-V3 (DeepSeek), Doubao-1.5-pro (Doubao), GPT-3.5 (GPT3.5), GPT-4.1 (GPT4.1), Moonshot V1 (Kimi), Mistral Large 2.1 (MistralL), Mistral Medium 3.1 (MistralM), Qwen-Plus (QwenP), and Qwen2.5-32B (Qwen2.5). We access models primarily via official APIs, except DeepSeek-V3, which we query through a third-party deployment due to official API discontinuation. We use the abbreviations in parentheses throughout the remainder of this section; exact model IDs are listed in Appendix A.5.

**Benchmarks.** We evaluate on AIME'24, MMLU, and GPQA: AIME for competition-level mathematical reasoning, MMLU for broad knowledge across 57 subjects, and GPQA for expert-validated science questions. MMLU and GPQA results are deferred to Appendix A.6 due to page limits. Our methodology is benchmark- and task-agnostic; we further validate generality on SWE-bench, MMBench, and Arena-Hard-v0.1 (Appendix A.7), covering software-engineering patch generation, multimodal reasoning, and challenging open-ended dialogue evaluation, respectively.

**Experimental Objectives.** Our experiments are designed to address three core targets: 1) Demonstrate the necessity of constructing SCS and A-SCS for LLM evaluation

by varying the settings of each indispensable component within SCS and A-SCS. 2) Decouple component effects and achieve precise attribution of performance variance using Analysis of Variance (ANOVA), thereby identifying the key components affecting LLM behavior. 3) Validate the superiority of LLM evaluatology by comparing it against traditional LLM evaluation methods, revealing its unique capability to obtain the performance ground truth.

**Evaluation Settings.** All experiments in Sections 5.2–5.4 are run under a hybrid single-thread and multi-thread evaluation setup to improve evaluation efficiency. Our automated evaluation tool can flexibly configure the number of testing threads through an input parameter.

### 5.2. The Necessity of Constructing SCS and A-SCS

This section reveals significant performance variations across SCS and A-SCS configurations, challenging the validity of traditional single-setting evaluations. To capture the performance variance, in the SCS experiments, we conducted 500 random samplings without replacement from the configuration space described in Section 4.1, with variables summarized in Table 2. For the A-SCS experiments, we utilized the augmented dataset constructed via the procedure in Section 4.3 and reused the identical 500 configurations from the SCS setup.

Under any given configuration, for each original question we form a candidate pool consisting of the original item and all of its augmented variants, and randomly sample exactly one instance from this pool for evaluation. The sampling scheme is designed such that the total probability of selecting either the original question or any analogical variant is $1/2$, and the total probability of selecting any novel variant is also $1/2$. This procedure ensures balanced coverage of the three dataset layers (seen, analogical, and novel) without exhaustively evaluating all variants of every question. Under this setup, we apply the same stopping rules in Section 4.4 to A-SCS and empirically find that accuracy estimates stabilize by 500 configurations, which we adopt as a conservative sample-size upper bound for SCS and A-SCS.

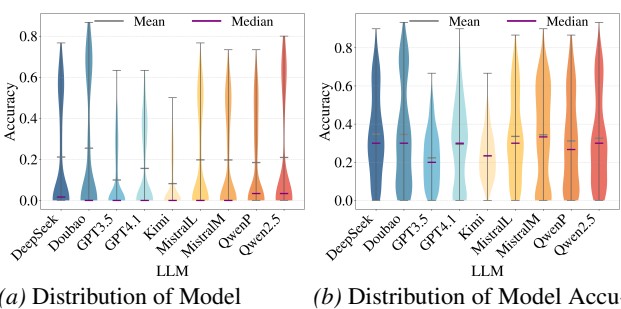

*(a)* Distribution of Model Accuracies on SCS

*(b)* Distribution of Model Accuracies on A-SCS

*Figure 5.* Distribution of Model Accuracies.

As shown in Figure 5, accuracy varies substantially across models and configuration spaces. Under SCS, most models attain peak accuracies between 0.6 and 0.8, with a small subset below 0.6; under A-SCS, all models exceed 0.6 at their peak, and most exceed 0.8. The concentration of near-zero accuracies in Figure 5a is attributable to the intrinsic difficulty of AIME questions rather than extreme value range settings of components; a detailed analysis is provided in Appendix A.3. Table 3 reports model rankings on the original, SCS, and A-SCS datasets; for SCS and A-SCS, rankings are by mean accuracy over sampled configurations, with ties at zero accuracy broken alphabetically by model name.

Drawing insights from the rankings, we observe three key conclusions: first, the original evaluation methodology demonstrates limited effectiveness in benchmarking large language models (LLMs) due to its inability to distinguish performance beyond two models achieving non-zero accuracy scores; second, DeepSeek ranks first on both Original and A-SCS and remains competitive under SCS (second), underscoring its robustness and superior generalization capabilities; third, model performance rankings exhibit contextual sensitivity, as evidenced by Doubao's inferior performance relative to DeepSeek in both Original and A-SCS datasets, yet its top-ranking achievement in SCS, thereby highlighting the non-transitive nature of LLM performance across varying data distributions.

### 5.3. Quantify the Contribution of Each Indispensable Component to Overall Performance Variance

In LLM evaluation, a key challenge is to decouple overall performance effects and accurately attribute performance variance to individual components illustrated in Fig. 1. Given the vast EC configuration space, exhaustive evaluation is computationally infeasible. To address this, we selected a limited number of experimental points from the full space, allowing us to systematically and evenly examine the effects of multiple components and their levels on performance with significantly fewer trials. This design reduces experimental cost while maintaining scientific rigor and representativeness.

To quantify the proportion of performance variance explained by each SCS component, we adopted an analysis of variance (ANOVA) approach. Specifically, for component $C_1$–$C_{10}$, we selected two levels ("high" and "low") within their respective ranges, with these ranges given in Table 2, thereby constructing a subspace of size $2^{10} = 1024$. For the Language component, we selected Chinese and English, while for three-valued components we used their maximum and minimum values. Within this subspace, variance decomposition was used to quantify the contributions of different components and their interactions to variations in accuracy. Moreover, we employed a permutation test to evaluate statistical significance, enabling a more robust assessment of

*Table 3.* Performance Rankings of LLMs.

| Type | 1 | 2 | 3 | 4 | 5 | 6 | 7 | 8 | 9 |
|------|---|---|---|---|---|---|---|---|---|
| Original | DeepSeek (0.367) | QwenP (0.333) | GPT4.1 (0.100) | Doubao (0.030) | GPT3.5 (0) | Kimi (0) | MistralL (0) | MistralM (0) | Qwen2.5 (0) |
| SCS | Doubao (0.254) | DeepSeek (0.211) | Qwen2.5 (0.209) | MistralL (0.197) | MistralM (0.197) | QwenP (0.184) | GPT4.1 (0.156) | GPT3.5 (0.100) | Kimi (0.082) |
| A-SCS | DeepSeek (0.347) | Doubao (0.346) | MistralM (0.344) | MistralL (0.336) | Qwen2.5 (0.326) | QwenP (0.312) | GPT4.1 (0.294) | Kimi (0.235) | GPT3.5 (0.223) |

*Note:* Models are sorted alphabetically by name when accuracy equals zero.

*Table 4.* ANOVA Results on DeepSeek (sorted by effect size in descending order).

| Factor | Effect Size $\eta^2$ | *p*-value |
|--------|------------------|---------|
| Question Format | 0.399643 | 0.000 |
| Question Format - CoT | 0.161394 | 0.000 |
| CoT | 0.080156 | 0.000 |
| max_tokens | 0.028099 | 0.000 |
| Question Format - Shot | 0.011101 | 0.000 |
| Language - Question Format | 0.008178 | 0.006 |
| CoT - max_tokens | 0.006721 | 0.010 |
| Language - CoT | 0.004345 | 0.038 |
| Multi Turn - max_tokens | 0.003841 | 0.050 |
| Language | 0.003841 | 0.046 |
| Shot - max_tokens | 0.003669 | 0.046 |
| Question Format - max_tokens | 0.002687 | 0.100 |
| Language - Multi Turn | 0.002600 | 0.066 |
| temperature - top_p | 0.002082 | 0.178 |
| Question Format - Multi Turn | 0.001321 | 0.244 |

component importance without relying on additional distributional assumptions. This procedure yields both the relative importance and the statistical significance of all components.

Taking the DeepSeek model as an example, Table 4 reports the main effects and two-way interactions that significantly influence its accuracy on the AIME benchmark, with the complete ANOVA results provided in Appendix A.2. Overall, Question Format, CoT, max_tokens, and their interactions with other components exhibit the most significant effects. Shot, Multi Turn, and Language also show significant effects, while the remaining components have only limited impact.

Consistent patterns were observed across other LLMs (see Appendix A.2). Using $p < 0.05$ as the significance threshold, we found that the main effects of Question Format and CoT, or their interactions with other components, were consistently significant across all LLMs. Furthermore, max_tokens, Shot, and Multi Turn also reached significance for the vast majority of models. In addition to these five core components, Language, top_p, and temperature were significant for some models. It is worth noting that for the remaining two components, Question Paraphrase and presence_penalty, the $p$-values did not meet the significance threshold, but reached **0.19** and **0.16**, respectively, on GPT4.1. This suggests that they may exert some influence on model performance, although the evidence is not sufficient for a definitive conclusion.

Since questions within A-SCS under the same EC configuration may be generated via different pipelines, it is no longer realistic to reuse the same ANOVA design as on SCS. Instead, we model the probability of answering each item

correctly with the ten EC components as fixed effects, while treating the specific question and augmentation pipeline as random effects. Consistent with the SCS results, CoT and Question Format remain the most influential fixed effects across models. In addition, max_tokens and Language are also statistically significant in all models, while Multi Turn and Shot, as in SCS, reach significance in the vast majority of models. The estimated random-intercept standard deviations for augmentation pipelines (0.87-1.03) are consistently larger than those for individual question instances (0.39-0.60) across all models. This indicates that, beyond question-level differences in difficulty, the choice of augmentation pipeline introduces substantial additional variability in the probability of answering an item correctly. Detailed per-model statistics are reported in Appendix A.8.

### 5.4. LLM Evaluatology vs. Traditional LLM Evaluation Methods

This section validates the superiority of LLM evaluatology by comparing it against traditional evaluation methods. While traditional methods rely on a single, static configuration, our LLM evaluatology uncovers the performance ground truth through comprehensive sampling.

Based on the randomly sampled data collected from the complete configuration space spanned by the components in Table 2, we estimated the overall average accuracies of different models on the same benchmark using their 95% and 99% confidence intervals. As illustrated in Figure 6a, we report the performance of different models on AIME, where the purple dots denote the test results under the commonly adopted default setting, using the original dataset without optimized prompting methods and with default decoding parameters. It can be observed that the purple dots are far from the confidence intervals (interval estimation of the population mean) obtained through random sampling, showing that evaluating a model under a single configuration is unreliable. Note that the accuracy values of 0 in Figure 6a are not due to missing data, but to the high difficulty of AIME questions, which are challenging even for human contestants.

Figure 6b further presents, on AIME, the accuracy differences between two models under the default configuration, together with the 95% and 99% confidence intervals constructed from accuracy differences observed across sampled

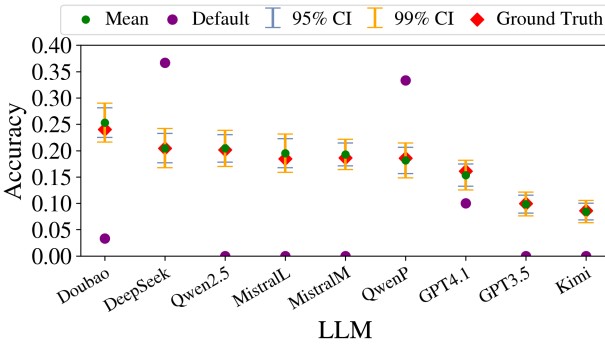

*(a)* CIs for LLM Accuracy on AIME

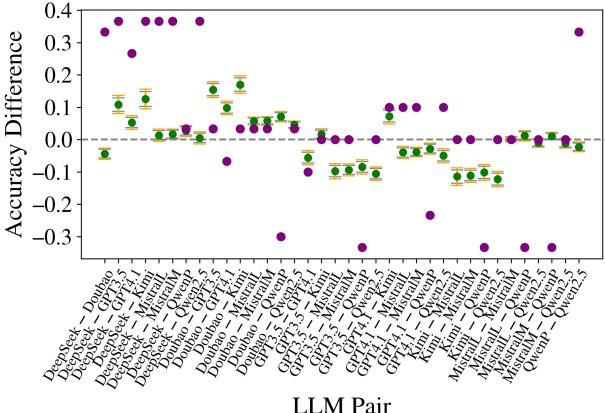

*(b)* CIs for Pairwise Accuracy Differences on AIME

*Figure 6.* Comparison of LLM Evaluatology and the Traditional Method on AIME.

equivalent evaluation configurations. We find three systematic mismatches between default-configuration comparisons and interval-based pairwise conclusions (18 distinct cases in total). (1) *Sign reversal* (9 cases): the default configuration favors one model, but the confidence interval for the accuracy difference supports the opposite ranking (e.g., Doubao vs. GPT4.1). (2) *Missed difference* (7 cases): the default configuration indicates no meaningful gap, yet the interval excludes zero, implying a significant ability difference (e.g., Kimi vs. MistralL). (3) *Illusory gap* (3 cases): the default configuration shows a sizable accuracy gap, yet the interval includes zero, so the models are statistically indistinguishable (e.g., DeepSeek vs. Qwen2.5). These discrepancies show that single-configuration testing is insufficient for claims of model superiority. Additional results on MMLU and GPQA are provided in Appendix A.6.

To further validate the reliability of our methodology in capturing the performance ground truth, we conducted a "restricted-space" verification. Leveraging the attribution results from Section 5.3, we identified the five most influential components for each LLM and constructed a focused configuration subspace. We then performed exhaustive testing within this subspace to derive a precise "restricted-space ground truth" (represented by the red diamonds in

Figure 6a). Crucially, for all evaluated models, this exhaustive ground truth falls consistently within the confidence intervals estimated by our random sampling approach. This alignment serves as strong empirical validation: it confirms that our LLM evaluatology does not merely estimate a range, but accurately bounds the intrinsic ground truth, demonstrating both validity and robustness compared to unstable single-configuration methods.

To assess the robustness of our sampling procedure, we re-run evaluation under a fixed, non-adaptive budget of 500 configurations per model—the conservative upper bound established above. Relative to the main-text results, the qualitative conclusions are unchanged: model rankings remain stable and main-effect patterns are essentially identical. Full results are reported in Appendix A.9.

### 5.5. Discussion

The methodology and framework are benchmark-agnostic and require no task-specific redesign. Extensions to SWE-bench, MMBench, and Arena-Hard subsets (Appendix A.7) show the same qualitative pattern: default-configuration results are poor proxies for the systematically sampled configuration distribution, indicating that the phenomenon extends to code, multimodal, and open-ended evaluation.

Our contribution is not merely quantitative—more components or larger score swings—but qualitative: joint modeling of ten EC components changes which evaluation conclusions remain stable. Findings from isolated-component studies often do not transfer once indispensable components are analyzed together, yielding component-specific scale effects, conditional rather than global recommendations, and a re-ranking of dominant drivers. Separating controllable evaluation variability from model capability thus alters not only the magnitude of observed differences, but the interpretive center of evaluatological claims.

### 6. Conclusion

LLM evaluatology establishes a principled framework for assessing LLMs by combining SCM with factorial decomposition under design of experiments (DOE), accounting for confounding evaluation components and separating model capability from evaluation-induced variability. Through a self-contained system (SCS) and its augmented extension (A-SCS), we make the EC configuration space explicit and show that meaningful evaluation requires joint consideration of dataset heterogeneity and systematic configuration exploration, not default-setting reporting alone. We advocate for the adoption of evaluatology as a foundational paradigm, encouraging the community to develop richer dataset augmentation strategies and robust evaluation practices that mirror the complexity of actual deployment scenarios.

## Impact Statement

This work introduces LLM evaluatology, a principled framework for large language model evaluation that enables systematic, causally informed analysis of model performance. By jointly modeling evaluation components and structured question variations, our framework provides a rigorous foundation for disentangling intrinsic model capabilities from confounding effects of the evaluation system. Beyond advancing methodological rigor in LLM benchmarking, this approach establishes a pathway for reproducible, interpretable, and causally faithful assessment across diverse conditions, offering the broader machine learning community a generalizable paradigm for principled evaluation. By enabling precise attribution of performance drivers, the framework can directly inform model design and optimization, guiding architecture choices, training strategies, and prompting methods. Its emphasis on holistic, system-level analysis further has potential impact on the development, deployment, and future benchmark design, supporting more trustworthy, reliable, and scientifically grounded insights into AI capabilities.

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

# A. Appendix

## A.1. Evaluation Settings on Different Benchmarks

*Table 5.* Evaluation Settings Reported in Technical Reports of Different LLMs. (Lang.=Language; Fmt.=Question Format; Para.=Question Paraphrase; CoT=chain-of-thought; MT.=Multi Turn; Temp.=temperature; PP=presence_penalty; MaxTok.=max_tokens; ori=original; Eng. = English; Mul. = Multiple language; y/n/x=yes/no/not reported).

*(a)* Evaluation Settings on AIME'2024

| Model | Dataset | | | Prompting | | | Decoding | | | |
|---|---|---|---|---|---|---|---|---|---|---|
| | Lang. | Fmt. | Para. | Shot | CoT | MT. | Temp. | top_p | PP | MaxTok. |
| DeepSeek-R1 | Eng. | ori | n | x | x | n | 0.6 | 0.95 | x | 32768 |
| DeepSeek-V3 | Eng. | ori | n | x | x | n | 0.7 | x | x | 8192 |
| Kimi K2 | Eng. | ori | n | x | n | n | 0.0 | x | x | 8192 |
| Kimi K1.5 | Eng. | ori | n | x | y | n | x | x | x | x |
| Qwen2 | | | | | Not evaluated on AIME | | | | | |
| Qwen2.5 | | | | | Not evaluated on AIME | | | | | |
| Qwen3 | Eng. | ori | n | x | y/n | n | 0.6/0.7 | 0.8/0.95 | 0.5/1.5 | 38912 |
| GPT-4 | | | | | Not evaluated on AIME | | | | | |
| GPT-4.1 | Eng. | ori | n | x | x | n | x | x | x | x |
| GPT-5 | | | | | Not evaluated on AIME | | | | | |
| Claude Opus 4 | | | | | Not evaluated on AIME | | | | | |
| Mistral Small 3.1 | | | | | Not evaluated on AIME | | | | | |
| Mistral Medium 3 | | | | | Not evaluated on AIME | | | | | |
| Mistral Large 2 | | | | | Not evaluated on AIME | | | | | |

*(b)* Evaluation Settings on MMLU

| Benchmark | Dataset | | | Prompting | | | Decoding | | | |
|---|---|---|---|---|---|---|---|---|---|---|
| | Lang. | Fmt. | Para. | Shot | CoT | MT. | Temp. | top_p | PP | MaxTok. |
| DeepSeek-R1 | Eng. | ori | n | 0 | y | n | 0.6 | 0.95 | x | 32768 |
| DeepSeek-V3 | Eng. | ori | n | 0 | y | n | x | x | x | 8192 |
| Kimi K2 | Eng. | ori | n | 5 | n | n | 0.0 | x | x | 8192 |
| Kimi K1.5 | Eng. | ori | n | x | y | n | x | x | x | x |
| Qwen2 | Eng. | ori | n | 5 | n | n | x | x | x | x |
| Qwen2.5 | Eng. | ori | n | 5 | n | n | x | x | x | x |
| Qwen3 | Mul. | ori | n | 5 | y/n | n | 0.6/0.7 | 0.8/0.95 | 0.5/1.5 | 32768 |
| GPT-4 | Mul. | ori | n | 5/3 | n | n | 0.3 | x | x | x |
| GPT-4.1 | Mul. | ori | n | x | x | n | x | x | x | x |
| GPT-5 | Mul. | ori | n | 0 | y/n | n | x | x | x | x |
| Claude Opus 4 | Mul. | ori | n | x | y/n | n | x | x | x | x |
| Mistral Small 3.1 | Eng. | ori | n | x | x | n | x | x | x | x |
| Mistral Medium 3 | | | | | Not evaluated on MMLU | | | | | |
| Mistral Large 2 | Mul. | ori | n | x | x | n | x | x | x | x |

*(c)* Evaluation Settings on GPQA

| Benchmark | Dataset | | | Prompting | | | Decoding | | | |
|---|---|---|---|---|---|---|---|---|---|---|
| | Lang. | Fmt. | Para. | Shot | CoT | MT. | Temp. | top_p | PP | MaxTok. |
| DeepSeek-R1 | Eng. | ori | n | 0 | y | n | 0.6 | 0.95 | x | 32768 |
| DeepSeek-V3 | Eng. | ori | n | 0 | y | n | x | x | x | 8192 |
| Kimi K2 | Eng. | ori | n | 5 | n | n | 0.0 | x | x | 8192 |
| Kimi K1.5 | | | | | Not evaluated on GPQA | | | | | |
| Qwen2 | Eng. | ori | n | 5 | n | n | x | x | x | x |
| Qwen2.5 | Eng. | ori | n | 5 | n | n | x | x | x | x |
| Qwen3 | Eng. | ori | n | 5 | y/n | n | 0.6/0.7 | 0.95/0.8 | 0.5/1.5 | 32768 |
| GPT-4 | | | | | Not evaluated on GPQA | | | | | |
| GPT-4.1 | Eng. | ori | n | x | x | n | x | x | x | x |
| GPT-5 | Eng. | ori | n | x | 0/1 | n | x | x | x | x |
| Claude Opus 4 | Eng. | ori | orn | x | 0/1 | n | x | x | x | x |
| Mistral Small 3.1 | Eng. | ori | n | x | x | n | x | x | x | x |
| Mistral Medium 3 | Eng. | ori | n | 5 | 1 | n | x | x | x | x |
| Mistral Large 2 | | | | | Not evaluated on GPQA | | | | | |

*(d)* Evaluation Settings on MATH

| Benchmark | Dataset | | | Prompting | | | Decoding | | | |
|---|---|---|---|---|---|---|---|---|---|---|
| | Lang. | Fmt. | Para. | Shot | CoT | MT. | Temp. | top_p | PP | MaxTok. |
| DeepSeek-R1 | Eng. | ori | n | 0/8 | n | n | 0.6 | 0.95 | x | 32768 |
| DeepSeek-V3 | Eng. | ori | n | 0/8 | n | n | 0.0 | x | x | 8192 |
| Kimi K2 | Eng. | ori | n | 4 | n | n | 0.0 | x | x | 8192 |
| Kimi K1.5 | Eng. | ori | n | x | y | n | x | x | x | x |
| Qwen2 | Eng. | ori | n | 4 | n | n | x | x | x | x |
| Qwen2.5 | Eng. | ori | n | 4 | n | n | x | x | x | x |
| Qwen3 | Eng. | ori | n | 4 | y/n | n | 0.6/0.7 | 0.8/0.95 | 0.5/1.5 | 32768 |
| GPT-4 | | | | Not evaluated on MATH | | | | | | |
| GPT-4.1 | | | | Not evaluated on MATH | | | | | | |
| GPT-5 | | | | Not evaluated on MATH | | | | | | |
| Claude Opus 4 | | | | Not evaluated on MATH | | | | | | |
| Mistral Small 3.1 | Eng. | ori | n | x | x | n | x | x | x | x |
| Mistral Medium 3 | Eng. | ori | n | 0 | n | n | x | x | x | x |
| Mistral Large 2 | Eng. | ori | n | 0 | n | n | x | x | x | x |

*(e)* Evaluation Settings on SWE-bench

| Benchmark | Dataset | | | Prompting | | | Decoding | | | |
|---|---|---|---|---|---|---|---|---|---|---|
| | Lang. | Fmt. | Para. | Shot | CoT | MT. | Temp. | top_p | PP | MaxTok. |
| DeepSeek-R1 | Eng. | ori | n | 0 | n | n | 0.6 | 0.95 | x | 32769 |
| DeepSeek-V3 | Eng. | ori | n | 0 | n | n | x | x | x | 8192 |
| Kimi K2 | Eng. | ori | n | x | n | n | 0.0 | x | x | 16384 |
| Kimi K1.5 | | | | Not evaluated on SWE-bench | | | | | | |
| Qwen2 | | | | Not evaluated on SWE-bench | | | | | | |
| Qwen2.5(pre) | | | | Not evaluated on SWE-bench | | | | | | |
| Qwen3(pre) | | | | Not evaluated on SWE-bench | | | | | | |
| GPT-4 | | | | Not evaluated on SWE-bench | | | | | | |
| GPT-4.1 | Eng. | ori | n | x | x | n | x | x | x | x |
| GPT-5 | Eng. | ori | n | x | y/n | n | x | x | x | x |
| Claude Opus 4 | Eng. | ori | n | x | y/n | n | x | 0.95 | x | x |
| Mistral Small 3.1 | | | | Not evaluated on SWE-bench | | | | | | |
| Mistral Medium 3 | | | | Not evaluated on SWE-bench | | | | | | |
| Mistral Large 2 | | | | Not evaluated on SWE-bench | | | | | | |

*(f)* Evaluation Settings on IFEval

| Benchmark | Dataset | | | Prompting | | | Decoding | | | |
|---|---|---|---|---|---|---|---|---|---|---|
| | Lang. | Fmt. | Para. | Shot | CoT | MT. | Temp. | top_p | PP | MaxTok. |
| DeepSeek-R1 | Eng. | ori | n | x | x | n | 0.6 | 0.95 | x | 32768 |
| DeepSeek-V3 | Eng. | ori | n | x | x | n | x | x | x | 8192 |
| Kimi K2 | Eng. | ori | n | x | n | n | 0.0 | x | x | 8192 |
| Kimi K1.5 | Eng. | ori | n | x | y | n | x | x | x | x |
| Qwen2 | Eng. | ori | n | x | x | n | x | x | x | x |
| Qwen2.5 | Eng. | ori | n | x | x | n | x | x | x | x |
| Qwen3 | Eng. | ori | n | x | y/n | n | 0.6/0.7 | 0.8/0.95 | 0.5/1.5 | 32768 |
| GPT-4 | | | | Not evaluated on IFEval | | | | | | |
| GPT-4.1 | Eng. | ori | n | x | x | n | x | x | x | x |
| GPT-5 | | | | Not evaluated on IFEval | | | | | | |
| Claude Opus 4 | | | | Not evaluated on IFEval | | | | | | |
| Mistral Small 3.1 | | | | Not evaluated on IFEval | | | | | | |
| Mistral Medium 3 | Eng. | ori | n | 0 | n | n | x | x | x | x |
| Mistral Large 2 | | | | Not evaluated on IFEval | | | | | | |

*(g)* Evaluation Settings on Arena-Hard

| | Dataset | | | Prompting | | | Decoding | | | |
|---|---|---|---|---|---|---|---|---|---|---|
| **Benchmark** | **Lang.** | **Fmt.** | **Para.** | **Shot** | **CoT** | **MT.** | **Temp.** | **top_p** | **PP** | **MaxTok.** |
| **DeepSeek-R1** | Eng. | ori | n | 0 | n | n | 0.6 | 0.95 | x | 32768 |
| **DeepSeek-V3** | Eng. | ori | n | 0 | n | n | x | x | x | 8192 |
| **Kimi K2** | Eng. | ori | n | x | n | n | 0.0 | x | x | 8192 |
| **Kimi K1.5** | | | | | Not evaluated on Arena-Hard | | | | | |
| **Qwen2** | Eng. | ori | n | x | x | n | x | x | x | x |
| **Qwen2.5** | Eng. | ori | n | x | x | n | x | x | x | x |
| **Qwen3** | Eng. | ori | n | x | y/n | n | 0.6/0.7 | 0.8/0.95 | 0.5/1.5 | 32768 |
| **GPT-4** | | | | | Not evaluated on Arena-Hard | | | | | |
| **GPT-4.1** | | | | | Not evaluated on Arena-Hard | | | | | |
| **GPT-5** | | | | | Not evaluated on Arena-Hard | | | | | |
| **Claude Opus 4** | | | | | Not evaluated on Arena-Hard | | | | | |
| **Mistral Small 3.1** | | | | | Not evaluated on Arena-Hard | | | | | |
| **Mistral Medium 3** | Eng. | ori | n | 0 | n | n | x | x | x | x |
| **Mistral Large 2** | Eng. | ori | n | x | x | n | x | x | x | x |

*(h)* Evaluation Settings on HumanEval

| | Dataset | | | Prompting | | | Decoding | | | |
|---|---|---|---|---|---|---|---|---|---|---|
| **Benchmark** | **Lang.** | **Fmt.** | **Para.** | **Shot** | **CoT** | **MT.** | **Temp.** | **top_p** | **PP** | **MaxTok.** |
| **DeepSeek-R1** | Eng. | ori | n | x | x | n | 0.6 | 0.95 | x | 32768 |
| **DeepSeek-V3** | Eng. | ori | n | x | x | n | x | x | x | 8192 |
| **Kimi K2** | Eng. | ori | n | x | x | n | x | x | x | x |
| **Kimi K1.5** | Eng. | ori | n | x | y | n | x | x | x | x |
| **Qwen2** | Eng. | ori | n | 0 | n | n | x | x | x | x |
| **Qwen2.5** | Eng. | ori | n | 0 | n | n | x | x | x | x |
| **Qwen3** | Eng. | ori | n | 0 | n | n | x | x | x | x |
| **GPT-4** | Eng. | ori | n | 0 | n | n | x | x | x | x |
| **GPT-4.1** | | | | | Not evaluated on HumanEval | | | | | |
| **GPT-5** | | | | | Not evaluated on HumanEval | | | | | |
| **Claude Opus 4** | | | | | Not evaluated on HumanEval | | | | | |
| **Mistral Small 3.1** | Eng. | ori | n | x | x | n | x | x | x | x |
| **Mistral Medium 3** | Eng. | ori | n | 0 | n | n | x | x | x | x |
| **Mistral Large 2** | Eng. | ori | n | x | x | n | x | x | x | x |

## A.2. ANOVA Analysis Results of SCS on Different LLMs

*Table 6.* Complete ANOVA Results on LLMs (sorted by effect size in descending order).

*(a)* Complete ANOVA Results on DeepSeek-V3

| Factor | Effect Size $\eta^2$ | $p$-value |
|---|---|---|
| **Question Format** | 0.399643 | 0.000 |
| **Question Format-CoT** | 0.161394 | 0.000 |
| **CoT** | 0.080156 | 0.000 |
| **max_tokens** | 0.028099 | 0.000 |
| **Question Format-Shot** | 0.011101 | 0.000 |
| **Language-Question Format** | 0.008178 | 0.006 |
| **CoT-max_tokens** | 0.006721 | 0.010 |
| **Language-CoT** | 0.004345 | 0.038 |
| **Multi Turn-max_tokens** | 0.003841 | 0.050 |
| **Language** | 0.003841 | 0.046 |
| **Shot-max_tokens** | 0.003669 | 0.046 |
| Question Format-max_tokens | 0.002687 | 0.100 |
| Language-Multi Turn | 0.002600 | 0.066 |
| temperature-top_p | 0.002082 | 0.178 |
| Question Format-Multi Turn | 0.001321 | 0.244 |
| Question Paraphrase | 0.001240 | 0.252 |
| Question Format-temperature | 0.001201 | 0.262 |
| Shot-temperature | 0.000926 | 0.326 |
| Multi Turn | 0.000793 | 0.364 |
| temperature | 0.000587 | 0.396 |
| CoT-Multi Turn | 0.000587 | 0.466 |
| CoT-top_p | 0.000573 | 0.482 |
| Language-Shot | 0.000534 | 0.466 |
| presence_penalty | 0.000435 | 0.488 |
| Question Format-Question Paraphrase | 0.000411 | 0.562 |
| Question Paraphrase-max_tokens | 0.000207 | 0.626 |
| Language-Question Paraphrase | 0.000198 | 0.660 |
| Shot-CoT | 0.000161 | 0.646 |
| CoT-temperature | 0.000140 | 0.698 |
| Language-max_tokens | 0.000140 | 0.712 |
| CoT-presence_penalty | 0.000140 | 0.668 |
| Shot | 0.000134 | 0.706 |
| Question Format-presence_penalty | 0.000133 | 0.708 |
| Shot-top_p | 0.000109 | 0.736 |
| max_tokens-presence_penalty | 0.000092 | 0.768 |
| max_tokens-top_p | 0.000086 | 0.790 |
| top_p | 0.000058 | 0.812 |
| Shot-Multi Turn | 0.000054 | 0.802 |
| temperature-max_tokens | 0.000038 | 0.836 |
| Language-presence_penalty | 0.000035 | 0.854 |
| top_p-presence_penalty | 0.000032 | 0.902 |
| Multi Turn-temperature | 0.000029 | 0.874 |
| Multi Turn-top_p | 0.000026 | 0.884 |
| Question Paraphrase-top_p | 0.000018 | 0.898 |
| temperature-presence_penalty | 0.000013 | 0.902 |
| Question Paraphrase-presence_penalty | 0.000011 | 0.910 |
| Language-top_p | 0.000008 | 0.942 |
| Question Paraphrase-Shot | 0.000006 | 0.930 |
| Multi Turn-presence_penalty | 0.000004 | 0.946 |
| Language-temperature | 0.000004 | 0.940 |
| Question Format-top_p | 0.000003 | 0.962 |
| Question Paraphrase-Multi Turn | 0.000003 | 0.972 |
| Shot-presence_penalty | 0.000001 | 0.966 |
| Question Paraphrase-CoT | 0.000001 | 0.996 |
| Question Paraphrase-temperature | 0.000000 | 0.994 |

*(b)* Complete ANOVA Results on Doubao-1.5-pro

| Factor | Effect Size $\eta^2$ | $p$-value |
|---|---|---|
| **Question Format** | 0.467626 | 0.000 |
| **Question Format-CoT** | 0.259657 | 0.000 |
| **CoT** | 0.130549 | 0.000 |
| **max_tokens** | 0.009192 | 0.002 |
| **CoT-max_tokens** | 0.008943 | 0.006 |
| max_tokens-presence_penalty | 0.000671 | 0.388 |
| Shot | 0.000638 | 0.408 |
| Shot-Multi Turn | 0.000605 | 0.424 |
| Question Paraphrase-max_tokens | 0.000502 | 0.466 |
| Multi Turn | 0.000473 | 0.466 |
| Multi Turn-max_tokens | 0.000464 | 0.472 |
| Language-temperature | 0.000455 | 0.468 |
| CoT-Multi Turn | 0.000427 | 0.496 |
| Question Format-Shot | 0.000400 | 0.546 |
| Question Format-max_tokens | 0.000392 | 0.538 |
| temperature | 0.000392 | 0.534 |
| Question Format-temperature | 0.000358 | 0.530 |
| Question Paraphrase-presence_penalty | 0.000349 | 0.552 |
| Language-top_p | 0.000326 | 0.584 |
| temperature-top_p | 0.000326 | 0.588 |
| Language-presence_penalty | 0.000310 | 0.578 |
| Language-Multi Turn | 0.000295 | 0.562 |
| Multi Turn-presence_penalty | 0.000272 | 0.604 |
| Language-CoT | 0.000265 | 0.586 |
| Shot-CoT | 0.000224 | 0.628 |
| Question Paraphrase-top_p | 0.000205 | 0.642 |
| Question Format-Multi Turn | 0.000158 | 0.704 |
| CoT-presence_penalty | 0.000147 | 0.696 |
| max_tokens-top_p | 0.000147 | 0.674 |
| presence_penalty | 0.000117 | 0.730 |
| temperature-max_tokens | 0.000108 | 0.740 |
| Question Format-presence_penalty | 0.000090 | 0.764 |
| Question Format-Question Paraphrase | 0.000067 | 0.792 |
| Question Paraphrase | 0.000054 | 0.836 |
| Shot-top_p | 0.000047 | 0.818 |
| Shot-presence_penalty | 0.000047 | 0.792 |
| Language | 0.000045 | 0.814 |
| Language-Question Paraphrase | 0.000044 | 0.840 |
| Multi Turn-temperature | 0.000036 | 0.832 |
| CoT-top_p | 0.000036 | 0.854 |
| CoT-temperature | 0.000034 | 0.834 |
| Language-Shot | 0.000034 | 0.850 |
| Shot-max_tokens | 0.000024 | 0.864 |
| temperature-presence_penalty | 0.000015 | 0.906 |
| top_p-presence_penalty | 0.000013 | 0.902 |
| Question Paraphrase-CoT | 0.000007 | 0.908 |
| Language-Question Format | 0.000002 | 0.956 |
| Question Paraphrase-temperature | 0.000001 | 0.960 |
| Shot-temperature | 0.000001 | 0.962 |
| Question Paraphrase-Multi Turn | 0.000001 | 0.966 |
| Language-max_tokens | 0.000001 | 0.970 |
| Question Paraphrase-Shot | 0.000001 | 0.980 |
| top_p | 0.000001 | 0.970 |
| Question Format-top_p | 0.000000 | 0.984 |
| Multi Turn-top_p | 0.000000 | 0.992 |

(c) Complete ANOVA Results on GPT-3.5

| Factor | Effect Size $\eta^2$ | $p$-value |
|---|---|---|
| **Question Format** | 0.417428 | 0.000 |
| **Question Format-CoT** | 0.199209 | 0.000 |
| **CoT** | 0.199208 | 0.000 |
| **temperature** | 0.006046 | 0.016 |
| **Question Format-temperature** | 0.005781 | 0.020 |
| **Shot-Multi Turn** | 0.005715 | 0.010 |
| **Shot-CoT** | 0.005651 | 0.026 |
| **max_tokens** | 0.005396 | 0.024 |
| **Question Format-max_tokens** | 0.004434 | 0.044 |
| temperature-top_p | 0.004320 | 0.052 |
| top_p | 0.003119 | 0.074 |
| Question Format-top_p | 0.002570 | 0.104 |
| CoT-max_tokens | 0.001773 | 0.170 |
| Question Format-Shot | 0.000853 | 0.344 |
| CoT-Multi Turn | 0.000828 | 0.352 |
| Language-Multi Turn | 0.000779 | 0.364 |
| Language | 0.000687 | 0.428 |
| Language-Question Format | 0.000600 | 0.484 |
| CoT-top_p | 0.000443 | 0.496 |
| CoT-temperature | 0.000425 | 0.508 |
| CoT-presence_penalty | 0.000408 | 0.508 |
| Question Paraphrase-Multi Turn | 0.000407 | 0.534 |
| Question Paraphrase-CoT | 0.000391 | 0.506 |
| Shot | 0.000296 | 0.578 |
| Shot-temperature | 0.000281 | 0.598 |
| Language-Shot | 0.000253 | 0.598 |
| max_tokens-presence_penalty | 0.000201 | 0.628 |
| Question Paraphrase-top_p | 0.000177 | 0.696 |
| Shot-max_tokens | 0.000135 | 0.720 |
| Language-top_p | 0.000115 | 0.690 |
| Question Paraphrase-presence_penalty | 0.000115 | 0.714 |
| Language-temperature | 0.000106 | 0.750 |
| Language-max_tokens | 0.000106 | 0.740 |
| Shot-presence_penalty | 0.000089 | 0.786 |
| Multi Turn-temperature | 0.000081 | 0.784 |
| Language-CoT | 0.000074 | 0.766 |
| Multi Turn-presence_penalty | 0.000074 | 0.788 |
| Question Format-Question Paraphrase | 0.000067 | 0.776 |
| Question Paraphrase | 0.000053 | 0.808 |
| Question Format-Multi Turn | 0.000047 | 0.826 |
| presence_penalty | 0.000047 | 0.854 |
| temperature-max_tokens | 0.000036 | 0.858 |
| Question Paraphrase-max_tokens | 0.000036 | 0.854 |
| temperature-presence_penalty | 0.000031 | 0.884 |
| Language-Question Paraphrase | 0.000027 | 0.858 |
| Question Format-presence_penalty | 0.000027 | 0.900 |
| Question Paraphrase-temperature | 0.000027 | 0.882 |
| Multi Turn | 0.000018 | 0.872 |
| Multi Turn-max_tokens | 0.000009 | 0.916 |
| Question Paraphrase-Shot | 0.000009 | 0.920 |
| Multi Turn-top_p | 0.000007 | 0.942 |
| top_p-presence_penalty | 0.000007 | 0.948 |
| Shot-top_p | 0.000007 | 0.942 |
| Language-presence_penalty | 0.000002 | 0.970 |
| max_tokens-top_p | 0.000000 | 0.978 |

(d) Complete ANOVA Results on GPT-4.1

| Factor | Effect Size $\eta^2$ | $p$-value |
|---|---|---|
| **Question Format** | 0.289086 | 0.000 |
| **Question Format-CoT** | 0.180162 | 0.000 |
| **CoT-max_tokens** | 0.054845 | 0.000 |
| **max_tokens** | 0.053399 | 0.000 |
| **CoT** | 0.027181 | 0.000 |
| **temperature-top_p** | 0.006685 | 0.010 |
| **Question Format-Shot** | 0.006529 | 0.030 |
| **temperature** | 0.005619 | 0.016 |
| **Shot** | 0.004963 | 0.028 |
| **max_tokens-top_p** | 0.004019 | 0.044 |
| Language-max_tokens | 0.003907 | 0.062 |
| Question Format-temperature | 0.003732 | 0.072 |
| top_p | 0.003364 | 0.098 |
| temperature-max_tokens | 0.002990 | 0.140 |
| Question Paraphrase-Shot | 0.002362 | 0.160 |
| Question Paraphrase-presence_penalty | 0.001939 | 0.194 |
| Shot-Multi Turn | 0.001842 | 0.194 |
| CoT-temperature | 0.001708 | 0.230 |
| Shot-CoT | 0.001589 | 0.290 |
| Language-CoT | 0.001517 | 0.246 |
| CoT-presence_penalty | 0.001406 | 0.276 |
| Question Format-max_tokens | 0.001360 | 0.306 |
| temperature-presence_penalty | 0.001284 | 0.292 |
| Language-Shot | 0.001276 | 0.262 |
| Language-Question Paraphrase | 0.001188 | 0.324 |
| presence_penalty | 0.001184 | 0.362 |
| CoT-top_p | 0.001183 | 0.300 |
| CoT-Multi Turn | 0.001037 | 0.376 |
| Multi Turn-max_tokens | 0.000988 | 0.376 |
| Shot-top_p | 0.000754 | 0.422 |
| Question Paraphrase-max_tokens | 0.000703 | 0.438 |
| Multi Turn-top_p | 0.000367 | 0.574 |
| Question Format-presence_penalty | 0.000348 | 0.584 |
| Question Format-Question Paraphrase | 0.000333 | 0.584 |
| Language-presence_penalty | 0.000329 | 0.586 |
| max_tokens-presence_penalty | 0.000324 | 0.600 |
| top_p-presence_penalty | 0.000253 | 0.618 |
| Shot-presence_penalty | 0.000192 | 0.702 |
| Shot-max_tokens | 0.000164 | 0.724 |
| Question Paraphrase | 0.000146 | 0.746 |
| Question Format-top_p | 0.000134 | 0.728 |
| Shot-temperature | 0.000133 | 0.724 |
| Question Format-Multi Turn | 0.000132 | 0.718 |
| Question Paraphrase-top_p | 0.000065 | 0.798 |
| Multi Turn-presence_penalty | 0.000055 | 0.828 |
| Language-top_p | 0.000023 | 0.896 |
| Language-Question Format | 0.000022 | 0.910 |
| Multi Turn-temperature | 0.000020 | 0.886 |
| Question Paraphrase-temperature | 0.000010 | 0.922 |
| Language-Multi Turn | 0.000009 | 0.906 |
| Question Paraphrase-CoT | 0.000009 | 0.922 |
| Question Paraphrase-Multi Turn | 0.000005 | 0.918 |
| Language | 0.000001 | 0.964 |
| Multi Turn | 0.000000 | 0.996 |
| Language-temperature | 0.000000 | 0.998 |

(e) Complete ANOVA Results on Qwen2.5-32B

| Factor | Effect Size $\eta^2$ | $p$-value |
|---|---|---|
| **Question Format** | 0.454352 | 0.000 |
| **Question Format-COT** | 0.204235 | 0.000 |
| **COT** | 0.200224 | 0.000 |
| **Question Format-Shot** | 0.009265 | 0.000 |
| **Shot** | 0.007983 | 0.008 |
| **Shot-COT** | 0.006715 | 0.002 |
| **Multi Turn** | 0.003717 | 0.046 |
| Question Format-Multi Turn | 0.003657 | 0.052 |
| max_tokens | 0.002764 | 0.080 |
| Language-Question Format | 0.002585 | 0.128 |
| COT-max_tokens | 0.001865 | 0.164 |
| Language | 0.001720 | 0.196 |
| COT-Multi Turn | 0.001540 | 0.214 |
| Multi Turn-max_tokens | 0.001110 | 0.322 |
| Question Format-max_tokens | 0.000922 | 0.292 |
| Language-COT | 0.000848 | 0.382 |
| Question Paraphrase-presence_penalty | 0.000710 | 0.400 |
| Language-Multi Turn | 0.000538 | 0.478 |
| temperature | 0.000430 | 0.444 |
| Shot-temperature | 0.000380 | 0.534 |
| Question Format-temperature | 0.000371 | 0.550 |
| Language-Question Paraphrase | 0.000343 | 0.550 |
| temperature-presence_penalty | 0.000334 | 0.566 |
| Language-temperature | 0.000290 | 0.602 |
| top_p | 0.000257 | 0.600 |
| max_tokens-top_p | 0.000242 | 0.618 |
| Shot-Multi Turn | 0.000234 | 0.654 |
| COT-temperature | 0.000219 | 0.660 |
| Question Format-Question Paraphrase | 0.000165 | 0.674 |
| Question Paraphrase-top_p | 0.000165 | 0.672 |
| Language-max_tokens | 0.000165 | 0.674 |
| Question Paraphrase | 0.000118 | 0.742 |
| Language-top_p | 0.000107 | 0.760 |
| temperature-top_p | 0.000107 | 0.724 |
| Multi Turn-presence_penalty | 0.000083 | 0.762 |
| max_tokens-presence_penalty | 0.000075 | 0.784 |
| Shot-max_tokens | 0.000062 | 0.770 |
| presence_penalty | 0.000051 | 0.830 |
| Question Format-top_p | 0.000038 | 0.824 |
| top_p-presence_penalty | 0.000038 | 0.830 |
| Multi Turn-temperature | 0.000029 | 0.856 |
| Question Paraphrase-temperature | 0.000027 | 0.882 |
| COT-presence_penalty | 0.000022 | 0.846 |
| Multi Turn-top_p | 0.000022 | 0.882 |
| Question Format-presence_penalty | 0.000022 | 0.866 |
| Language-presence_penalty | 0.000012 | 0.912 |
| temperature-max_tokens | 0.000012 | 0.918 |
| Question Paraphrase-Multi Turn | 0.000009 | 0.942 |
| Language-Shot | 0.000007 | 0.926 |
| Question Paraphrase-Shot | 0.000005 | 0.928 |
| COT-top_p | 0.000003 | 0.954 |
| Shot-presence_penalty | 0.000002 | 0.960 |
| Shot-top_p | 0.000001 | 0.982 |
| Question Paraphrase-COT | 0.000000 | 0.984 |
| Question Paraphrase-max_tokens | 0.000000 | 0.990 |

(f) Complete ANOVA Results on Qwen-Plus

| Factor | Effect Size $\eta^2$ | $p$-value |
|---|---|---|
| **Question Format** | 0.302717 | 0.000 |
| **Question Format-COT** | 0.259678 | 0.000 |
| **COT** | 0.098747 | 0.000 |
| **Shot** | 0.047447 | 0.000 |
| **Question Format-Shot** | 0.042448 | 0.000 |
| **Shot-COT** | 0.024956 | 0.000 |
| **COT-max_tokens** | 0.016596 | 0.000 |
| **max_tokens** | 0.016280 | 0.000 |
| **Language** | 0.005019 | 0.024 |
| Language-Question Format | 0.003272 | 0.060 |
| temperature-top_p | 0.002324 | 0.112 |
| Multi Turn-max_tokens | 0.001979 | 0.154 |
| Question Format-temperature | 0.001662 | 0.202 |
| top_p | 0.001282 | 0.250 |
| Language-Shot | 0.000991 | 0.296 |
| Question Paraphrase-Multi Turn | 0.000877 | 0.320 |
| COT-temperature | 0.000822 | 0.368 |
| Multi Turn | 0.000736 | 0.428 |
| Question Format-Question Paraphrase | 0.000654 | 0.458 |
| temperature | 0.000608 | 0.412 |
| Language-COT | 0.000519 | 0.472 |
| Shot-Multi Turn | 0.000438 | 0.512 |
| Multi Turn-presence_penalty | 0.000400 | 0.512 |
| Shot-presence_penalty | 0.000375 | 0.496 |
| Question Format-top_p | 0.000275 | 0.604 |
| COT-Multi Turn | 0.000245 | 0.612 |
| Question Format-max_tokens | 0.000226 | 0.648 |
| Question Format-Multi Turn | 0.000166 | 0.708 |
| Multi Turn-temperature | 0.000135 | 0.724 |
| max_tokens-presence_penalty | 0.000128 | 0.690 |
| top_p-presence_penalty | 0.000102 | 0.752 |
| Question Paraphrase-Shot | 0.000101 | 0.746 |
| COT-top_p | 0.000090 | 0.758 |
| Language-presence_penalty | 0.000084 | 0.792 |
| Language-max_tokens | 0.000073 | 0.762 |
| temperature-presence_penalty | 0.000058 | 0.786 |
| Language-Multi Turn | 0.000058 | 0.806 |
| Question Paraphrase-top_p | 0.000058 | 0.814 |
| Language-temperature | 0.000049 | 0.816 |
| Shot-top_p | 0.000037 | 0.838 |
| Question Paraphrase | 0.000033 | 0.864 |
| Shot-temperature | 0.000033 | 0.860 |
| Language-top_p | 0.000029 | 0.850 |
| Question Paraphrase-max_tokens | 0.000023 | 0.862 |
| Question Paraphrase-presence_penalty | 0.000015 | 0.902 |
| Question Paraphrase-temperature | 0.000015 | 0.904 |
| Multi Turn-top_p | 0.000015 | 0.924 |
| Shot-max_tokens | 0.000013 | 0.898 |
| COT-presence_penalty | 0.000013 | 0.918 |
| Question Format-presence_penalty | 0.000005 | 0.946 |
| temperature-max_tokens | 0.000002 | 0.970 |
| presence_penalty | 0.000002 | 0.960 |
| max_tokens-top_p | 0.000002 | 0.970 |
| Question Paraphrase-COT | 0.000001 | 0.978 |
| Language-Question Paraphrase | 0.000000 | 0.994 |

### (g) Complete ANOVA Results on Mistral Large 2.1

| Factor | Effect Size $\eta^2$ | $p$-value |
|---|---|---|
| **Question Format** | 0.406873 | 0.000 |
| **Question Format-CoT** | 0.268919 | 0.000 |
| **CoT** | 0.075852 | 0.000 |
| **CoT-max_tokens** | 0.026017 | 0.000 |
| **max_tokens** | 0.025372 | 0.000 |
| **Question Format-Multi Turn** | 0.004796 | 0.024 |
| Multi Turn | 0.003609 | 0.058 |
| Question Format-Shot | 0.003338 | 0.068 |
| CoT-Multi Turn | 0.002708 | 0.100 |
| Question Format-max_tokens | 0.001379 | 0.230 |
| Shot | 0.001023 | 0.320 |
| Language-Question Format | 0.001004 | 0.280 |
| Multi Turn-max_tokens | 0.000881 | 0.310 |
| Shot-Multi Turn | 0.000830 | 0.368 |
| Language-Multi Turn | 0.000766 | 0.372 |
| Language-CoT | 0.000765 | 0.356 |
| Question Format-Question Paraphrase | 0.000644 | 0.414 |
| Multi Turn-presence_penalty | 0.000456 | 0.458 |
| Question Paraphrase | 0.000364 | 0.550 |
| Question Paraphrase-presence_penalty | 0.000353 | 0.486 |
| max_tokens-presence_penalty | 0.000272 | 0.620 |
| Shot-top_p | 0.000244 | 0.638 |
| Multi Turn-top_p | 0.000244 | 0.570 |
| Language-Question Paraphrase | 0.000235 | 0.618 |
| Question Format-presence_penalty | 0.000227 | 0.604 |
| CoT-temperature | 0.000227 | 0.628 |
| Question Paraphrase-Shot | 0.000193 | 0.658 |
| temperature | 0.000185 | 0.644 |
| Shot-presence_penalty | 0.000178 | 0.650 |
| Question Format-temperature | 0.000178 | 0.672 |
| Question Paraphrase-top_p | 0.000163 | 0.686 |
| Shot-max_tokens | 0.000163 | 0.682 |
| top_p | 0.000142 | 0.690 |
| Question Paraphrase-max_tokens | 0.000135 | 0.688 |
| Question Paraphrase-Multi Turn | 0.000128 | 0.772 |
| max_tokens-top_p | 0.000110 | 0.712 |
| Language | 0.000109 | 0.736 |
| CoT-presence_penalty | 0.000087 | 0.780 |
| Shot-temperature | 0.000087 | 0.810 |
| Question Paraphrase-CoT | 0.000053 | 0.818 |
| top_p-presence_penalty | 0.000049 | 0.824 |
| presence_penalty | 0.000038 | 0.836 |
| Language-temperature | 0.000035 | 0.840 |
| Language-top_p | 0.000028 | 0.864 |
| temperature-max_tokens | 0.000017 | 0.884 |
| temperature-presence_penalty | 0.000015 | 0.884 |
| Multi Turn-temperature | 0.000007 | 0.928 |
| temperature-top_p | 0.000006 | 0.928 |
| Question Paraphrase-temperature | 0.000005 | 0.950 |
| CoT-top_p | 0.000003 | 0.952 |
| Language-Shot | 0.000003 | 0.944 |
| Question Format-top_p | 0.000002 | 0.952 |
| Language-max_tokens | 0.000002 | 0.950 |
| Language-presence_penalty | 0.000002 | 0.960 |
| Shot-CoT | 0.000000 | 0.998 |

### (h) Complete ANOVA Results on Mistral Medium 3.1

| Factor | Effect Size $\eta^2$ | $p$-value |
|---|---|---|
| **Question Format** | 0.430500 | 0.000 |
| **Question Format-CoT** | 0.274248 | 0.000 |
| **CoT** | 0.105919 | 0.000 |
| **CoT-max_tokens** | 0.013038 | 0.002 |
| **max_tokens** | 0.012064 | 0.000 |
| **Question Format-Multi Turn** | 0.007865 | 0.004 |
| **Multi Turn** | 0.005334 | 0.026 |
| Shot | 0.003097 | 0.064 |
| CoT-Multi Turn | 0.003001 | 0.090 |
| Question Format-Shot | 0.002782 | 0.086 |
| Multi Turn-max_tokens | 0.001354 | 0.252 |
| Shot-Multi Turn | 0.000908 | 0.336 |
| Language-Question Paraphrase | 0.000891 | 0.378 |
| Shot-CoT | 0.000636 | 0.434 |
| Question Paraphrase-top_p | 0.000366 | 0.476 |
| Question Paraphrase-presence_penalty | 0.000312 | 0.534 |
| Question Paraphrase-Multi Turn | 0.000282 | 0.578 |
| Language | 0.000254 | 0.626 |
| max_tokens-presence_penalty | 0.000227 | 0.650 |
| Language-Multi Turn | 0.000219 | 0.656 |
| Multi Turn-presence_penalty | 0.000218 | 0.642 |
| Question Format-temperature | 0.000178 | 0.682 |
| Multi Turn-top_p | 0.000163 | 0.672 |
| Language-CoT | 0.000163 | 0.690 |
| temperature-top_p | 0.000148 | 0.698 |
| Question Format-max_tokens | 0.000128 | 0.672 |
| Question Format-Question Paraphrase | 0.000109 | 0.746 |
| Question Paraphrase-max_tokens | 0.000103 | 0.754 |
| CoT-top_p | 0.000091 | 0.750 |
| CoT-temperature | 0.000081 | 0.764 |
| Shot-temperature | 0.000076 | 0.806 |
| top_p | 0.000076 | 0.784 |
| Question Paraphrase-temperature | 0.000076 | 0.758 |
| Language-temperature | 0.000061 | 0.798 |
| Question Format-presence_penalty | 0.000053 | 0.806 |
| Shot-top_p | 0.000048 | 0.808 |
| Language-Shot | 0.000044 | 0.832 |
| Shot-max_tokens | 0.000044 | 0.814 |
| Question Paraphrase-CoT | 0.000037 | 0.854 |
| presence_penalty | 0.000037 | 0.844 |
| Question Paraphrase | 0.000034 | 0.850 |
| Language-top_p | 0.000024 | 0.870 |
| Language-presence_penalty | 0.000022 | 0.874 |
| Shot-presence_penalty | 0.000019 | 0.888 |
| temperature-presence_penalty | 0.000014 | 0.910 |
| Language-max_tokens | 0.000008 | 0.920 |
| Language-Question Format | 0.000007 | 0.938 |
| Question Format-top_p | 0.000005 | 0.942 |
| temperature | 0.000005 | 0.948 |
| max_tokens-top_p | 0.000004 | 0.958 |
| Question Paraphrase-Shot | 0.000003 | 0.956 |
| Multi Turn-temperature | 0.000001 | 0.972 |
| temperature-max_tokens | 0.000001 | 0.968 |
| CoT-presence_penalty | 0.000000 | 0.986 |
| top_p-presence_penalty | 0.000000 | 0.994 |

(i) Complete ANOVA Results on Moonshot V1

| Factor | Effect Size $\eta^2$ | $p$-value |
|---|---|---|
| **Question Format** | 0.297180 | 0.000 |
| **Question Format-CoT** | 0.147641 | 0.000 |
| **CoT** | 0.130758 | 0.000 |
| **Shot-CoT** | 0.077565 | 0.000 |
| **Question Format-Shot** | 0.042631 | 0.000 |
| **Shot** | 0.038198 | 0.000 |
| **CoT-Multi Turn** | 0.018078 | 0.000 |
| **Language-Shot** | 0.005572 | 0.016 |
| CoT-max_tokens | 0.002887 | 0.090 |
| Language-Multi Turn | 0.002824 | 0.084 |
| Language | 0.002702 | 0.110 |
| Question Format-Multi Turn | 0.002700 | 0.106 |
| max_tokens | 0.002409 | 0.118 |
| Language-Question Format | 0.002297 | 0.108 |
| Multi Turn | 0.002185 | 0.124 |
| Multi Turn-max_tokens | 0.001677 | 0.166 |
| Question Format-max_tokens | 0.000966 | 0.300 |
| Language-max_tokens | 0.000930 | 0.356 |
| Shot-max_tokens | 0.000861 | 0.380 |
| Language-Question Paraphrase | 0.000793 | 0.360 |
| CoT-presence_penalty | 0.000402 | 0.494 |
| top_p-presence_penalty | 0.000314 | 0.554 |
| Multi Turn-temperature | 0.000274 | 0.578 |
| max_tokens-top_p | 0.000219 | 0.662 |
| Shot-presence_penalty | 0.000203 | 0.620 |
| top_p | 0.000142 | 0.694 |
| Question Format-top_p | 0.000128 | 0.742 |
| Question Paraphrase-max_tokens | 0.000128 | 0.724 |
| Question Paraphrase-CoT | 0.000115 | 0.756 |
| Shot-temperature | 0.000092 | 0.786 |
| max_tokens-presence_penalty | 0.000081 | 0.770 |
| Shot-top_p | 0.000081 | 0.792 |
| temperature-top_p | 0.000081 | 0.776 |
| Question Paraphrase-Shot | 0.000081 | 0.804 |
| Language-temperature | 0.000081 | 0.782 |
| temperature-presence_penalty | 0.000053 | 0.860 |
| Question Paraphrase-Multi Turn | 0.000053 | 0.858 |
| Shot-Multi Turn | 0.000045 | 0.812 |
| Question Format-temperature | 0.000037 | 0.856 |
| CoT-top_p | 0.000037 | 0.866 |
| Question Paraphrase-presence_penalty | 0.000037 | 0.882 |
| Multi Turn-presence_penalty | 0.000037 | 0.872 |
| Question Format-presence_penalty | 0.000030 | 0.878 |
| Question Paraphrase | 0.000024 | 0.864 |
| Language-presence_penalty | 0.000019 | 0.904 |
| Language-CoT | 0.000019 | 0.884 |
| Multi Turn-top_p | 0.000014 | 0.922 |
| temperature-max_tokens | 0.000002 | 0.958 |
| CoT-temperature | 0.000002 | 0.954 |
| Question Format-Question Paraphrase | 0.000002 | 0.968 |
| Question Paraphrase-temperature | 0.000001 | 0.966 |
| temperature | 0.000001 | 0.970 |
| Language-top_p | 0.000001 | 0.968 |
| presence_penalty | 0.000000 | 0.992 |
| Question Paraphrase-top_p | 0.000000 | 0.998 |

## A.3. Justification of SCS Component Value Ranges

The value ranges in Table 2 are the result of a trade-off between realism and tractability in a high-dimensional configuration space. This section provides a concise justification of these choices and their empirical consequences.

### (1) Coarsening of components grids.

In the underlying experiments, decoding parameters were initially explored on finer-grained grids. For the final SCS configuration space, these grids were deliberately coarsened (for example, to $\{0.0, 1.0, 2.0\}$ for temperature) to keep the 10-dimensional configuration space tractable: every additional level per variable multiplies the total number of configurations and makes systematic sampling quickly intractable.

The coarse grids in Table 2 should thus be viewed as a compressed parametrization of a richer underlying search. Empirically, runs on finer-grained grids yield qualitative trends that are consistent with those reported in the main paper.

### (2) `max_tokens = 10` does not truncate answers in our setup.

The configuration `max_tokens` $= 10$ is realistic and non-pathological in the specific setting considered here. The evaluation protocol for AIME-style problems requires the model to answer in a very compact, machine-parsable format:

- Multiple-choice: `####A####`

- Numeric fill-in-the-blank: `####342####`

Here `####` is a special delimiter that corresponds to a single token in the tokenizer; a complete answer such as `####A####` occupies only a few tokens (typically around 3 tokens). Under this constrained answer format, `max_tokens` $= 10$ is more than sufficient to produce a full valid answer for both multiple-choice and fill-in-the-blank questions.

**(3) `temperature = 2.0` as an aggressive but meaningful setting.**

While `temperature = 2.0` introduces aggressive sampling dynamics, it only creates the potential for random or uninformative outputs rather than mandating them. For instance, under the following configuration for DeepSeek-V3:

```
Language: yy
Question Format: 0
Question Paraphrase: 0
Shot: 1
CoT: 0
Multi Turn: 0
temperature: 2.0
top_p: 0.6
presence_penalty: 0.5
max_tokens: 100
```

the observed accuracy on AIME is 56.67%, which is far above random chance. This demonstrates that including `temperature = 2.0` in SCS does not simply inject invalid runs. Rather, `temperature = 2.0` is challenging but not pathological for the model, and it provides useful information about robustness under aggressive decoding settings.

**(4) Realistic deployment scenarios for "extreme" settings.**

`max_tokens = 10` does have practical use cases in real deployments, especially when users only need very short responses from the model. By constraining `max_tokens`, applications can cap output length to save time (avoiding long generations when only a brief signal is required) and reduce cost (since pricing is typically proportional to the number of output tokens). Consequently, although `max_tokens = 10` may appear "extreme" from the perspective of open-ended chat, it is a realistic and meaningful setting for short-answer tasks like ours, as well as for many production scenarios that prioritize brevity and efficiency.

Similarly, `temperature = 2.0` is not purely an academic extreme. It is used in creative-generation scenarios such as poetry, fiction, and brainstorming, where diversity and novelty are prioritized. It is also employed for generating unusual phrasing or surprising ideas in exploratory ideation tools, where users explicitly trade reliability for creativity. In such applications, practitioners intentionally set a high temperature to push the model away from generic responses. Therefore, `temperature = 2.0` can be regarded as a realistic configuration for specific use cases, even if it is not ideal for strict QA-type benchmarks. Including such settings makes it possible to study how sensitive model performance is to decoding extremes, reflecting the fact that real deployments often explore a wide range of temperatures across tasks.

**(5) Ablation excluding "extreme" settings.**

To assess the influence of these extreme-looking configurations on the overall accuracy distributions, an ablation was conducted in which all evaluation points satisfying `max_tokens = 10`, or `temperature = 2.0` were removed. The violin plots corresponding to Figure 5a were then recomputed and replotted. The resulting plots are qualitatively very similar to the originals: the density near zero accuracy is slightly reduced, but the overall shape remains. This indicates that the concentration near zero is not primarily driven by these "extreme" settings. Instead, it arises from a combination of the intrinsic difficulty of AIME, non-CoT configurations, less favorable prompts and languages, as well as other factors.

**A.4. A-SCS Construction Pipeline**

**Analogical: Distractor Insertion**

For distractor insertion, we define three explicit, controllable categories of redundancy and implement all instances via LLM prompting. To ensure that the inserted distractors strictly follow our predefined specifications, we empirically test several candidate LLMs and choose the one that most consistently adheres to these constraints (GPT-5). This selection is made solely to guarantee transformation fidelity rather than to compare model capabilities. For each item to be transformed, the chosen LLM is invoked through an API and, guided by our structured prompts, automatically produces and inserts the required redundant content. The concrete implementation is as follows.

(1) Context-irrelevant redundancy.

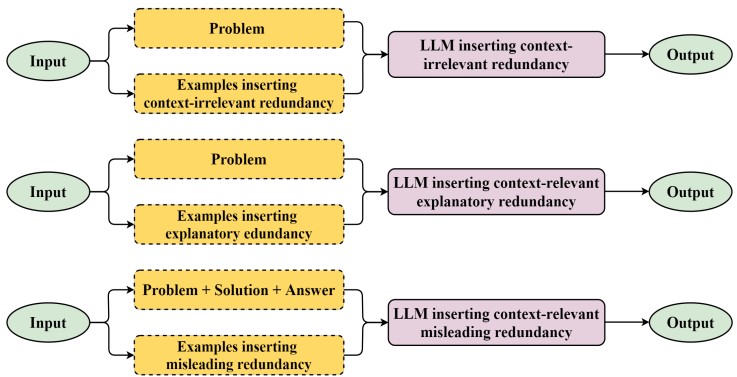

*Figure 7.* Distractor Insertion Flow Chart.

- Provide the LLM with an example containing an original question and a version with added context-irrelevant redundancy.
- Instruct the LLM to insert unrelated sentences at random positions, ensuring they are completely irrelevant to the target question.

---

**Algorithm 1** Context-irrelevant redundancy insertion

---

1: **function** INSERTCONTEXTIRRELEVANTDISTRACTOR(*question_text*)
2:     *EXAMPLE_PAIR* ← (*example_question, example_question_with_irrelevant_context*)
3:     *PROMPT* ← BUILDPROMPTIRRELEVANT(*example_pair = EXAMPLE_PAIR, target_problem = question_text*)
4:     *RESPONSE* ← LLM_CALL(*PROMPT*)
5:     *transformed_text* ← PARSETRANSFORMEDPROBLEM(*RESPONSE*)
6:     **return** *transformed_text*
7: **end function**

---

(2) Context-relevant, explanatory redundancy.

- Provide the LLM with an example of an original question and a version with added explanatory redundancy.
- Instruct the LLM to insert a redundant sentence at a random position in each target question that explains a concept already appearing in the target question.

---

**Algorithm 2** Context-relevant explanatory redundancy insertion

---

1: **function** INSERTEXPLANATORYDISTRACTOR(*question_text*)
2:     *EXAMPLE_PAIR* ← (*example_question, example_question_with_explanatory_sentence*)
3:     *PROMPT* ← BUILDPROMPTEXPLANATORY(*example_pair = EXAMPLE_PAIR, target_problem = question_text*)
4:     *RESPONSE* ← LLM_CALL(*PROMPT*)
5:     *transformed_text* ← PARSETRANSFORMEDPROBLEM(*RESPONSE*)
6:     **return** *transformed_text*
7: **end function**

---

(3) Context-relevant, misleading redundancy.

- Provide the LLM with an example containing an original question and a version with added misleading but logically related redundancy.
- Supply the model with the correct answer and several correct solution approaches, and instruct it to avoid directly hinting at these correct strategies when crafting the misleading cue. The official answer and solution approaches are provided by the user, and providing solution approaches is optional.
- Instruct the model to insert a redundant sentence that nudges the reader toward an incorrect strategy or line of reasoning, without explicitly revealing that it is "misleading" or "distracting".

---

**Algorithm 3** Context-relevant misleading redundancy insertion

---

1: **function** INSERTMISLEADINGDISTRACTOR(*question_text*, *answer_gold*, *solution_sketches*)
2:     *EXAMPLE_PAIR* ← (*example_question*, *example_question_with_misleading_sentence*)
3:     *PROMPT* ← BUILDPROMPTMISLEADING(*example_pair* = *EXAMPLE_PAIR*, *target_problem* = *question_text*, *answer_gold* = *answer_gold*, *solution_sketches* = *solution_sketches*)
4:     *RESPONSE* ← LLM_CALL(*PROMPT*)
5:     *transformed_text* ← PARSETRANSFORMEDPROBLEM(*RESPONSE*)
6:     **return** *transformed_text*
7: **end function**

---

In practice, the selected LLM produces variations that are more diverse and linguistically natural than manual editing. In particular, its context-relevant misleading redundancies tend to hint at incorrect heuristics in a more subtle way than hand-written versions, while still strictly adhering to the predefined category constraints. The entire process involves no per-item manual editing. The three examples of redundancy for three types generated by the above procedure are illustrated as follows:

1. Context-irrelevant redundancy example:

   *The weather today seems quite pleasant, and it might be a great day for a picnic.* Find the number of triples of nonnegative integers $(a, b, c)$ satisfying $a + b + c = 300$ and $a^2b + a^2c + b^2a + b^2c + c^2a + c^2b = 6{,}000{,}000$.

Here, the weather is entirely unrelated to the math content.

2. Context-relevant, explanatory redundancy example:

   There exist real numbers $x$ and $y$, both greater than 1, such that $\log_x(y^x) = \log_y(x^{4y}) = 10$. *A logarithm is a way to express how many times a base must be multiplied by itself to get a certain number.* Find $xy$.

The added sentence explains the notion of a logarithm while leaving the underlying problem unchanged.

3. Context-relevant, misleading redundancy example:

   Alice and Bob play the following game. A stack of $n$ tokens lies before them. The players take turns with Alice going first. On each turn, the player removes either 1 token or 4 tokens from the stack. *Many players adopt a greedy approach here: always take* 4 *whenever possible to shorten the game and restrict the opponent's replies.* Whoever removes the last token wins. Find the number of positive integers $n$ less than or equal to 2024 for which there exists a strategy for Bob that guarantees that Bob will win the game regardless of Alice's play.

The extra sentence about the "greedy approach" is logically related to the game but suggests a flawed strategy, intentionally nudging the solver toward an incorrect line of reasoning.

**Analogical: Numeric Substitutions**

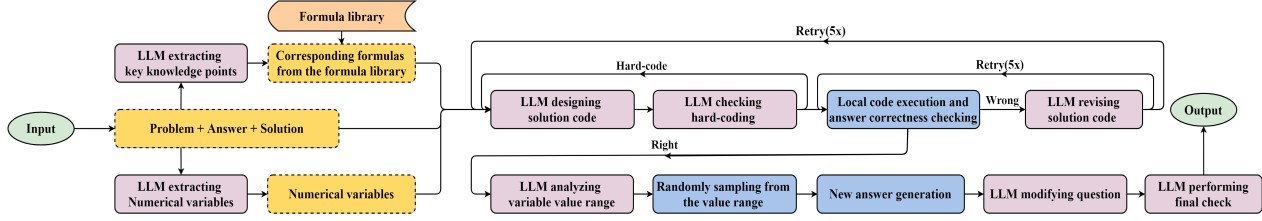

*Figure 8.* Numeric Substitutions Flow Chart.

For numeric substitutions, we use a uniform pipeline built around LLM-generated Python solvers and automatic verification scripts, rather than manually changing a few numbers:

- Invoke the LLM to extract the primary knowledge points tested by the problem.
- Query a pre-constructed formula library indexed by knowledge point to retrieve relevant formulas.
- Invoke the LLM to select a problem variable that is suitable for modification.
- Provide the LLM with the problem statement, the answer, the retrieved formulas, the variable, and (optionally) multiple solution sketches and prompt it to:
  - analyze the solution strategy, leveraging the provided solution sketches when available;
  - write a Python code where problem-specific numeric quantities exposed as explicit variables.
- Invoke the LLM to verify that the generated Python code implements a general computational procedure for solving the problem, rather than relying on hard-coded instance-specific outputs or trivial pattern matching.
- Write the generated code to a Python script file and execute it in a subprocess, call `solve()` with the variable set to its original value, and verify that the output matches the answer.
- If the code fails (wrong answer or runtime error), return the error message or incorrect output to the LLM and ask it to refine the code; repeat this refinement–verification loop for up to five attempts, and keep the code only if it passes on the original instance.
- Invoke the LLM to determine a reasonable value range for the selected variable.
- Sample a new value within the validated ranges.
- Run the generated code on the modified value to obtain the corresponding new answer.
- Invoke the LLM to generate variants of the origin problem.
- Invoke the LLM to conduct a final check of the entire pipeline and, if the final check passes, output the new problem and its corresponding answer.

---

**Algorithm 4** Numeric substitution

---

1: **function** GENERATENUMERICSUBSTITUTIONVARIANT(*problem_text*, *answer_gold*, *solution_sketches*, *max_iter*= 5, *max_refine*= 5)
2:     *knowledge_points* ← LLM_EXTRACT_KNOWLEDGE_POINTS(*problem_text*, *solution_sketches*)
3:     *retrieved_formulas* ← RETRIEVEFORMULAS(*knowledge_points*)
4:     *numeric_var* ← LLM_EXTRACT_NUMERIC_VARIABLE(*problem_text*)
5:     *history* ← EMPTY_LIST()
6:     *solver_code* ← FAILURE
7:     *value_ranges* ← NONE
8:     **for** *iter* = 1 **to** *max_iter* **do**
9:         *PROMPT* ← BUILDPROMPTCODEGEN(*problem_text*, *answer_gold*, *solution_sketches*, *retrieved_formulas*, *numeric_var*)
10:         APPEND(*history*, (*PROMPT*, NONE))
11:         *CODE* ← LLM_CALL(*PROMPT*)
12:         *is_hard_code* ← LLM_HARD_CODE_CHECK(*CODE*)
13:         **if** *is_hard_code* **then**
14:             **continue**
15:         **end if**
16:         **for** *refine_step* = 0 **to** *max_refine* **do**
17:             (*output*, *error*) ← RUNPYTHON(*CODE*, input = *numeric_var.value*)
18:             APPEND(*history*, (*CODE*, (*output*, *error*)))
19:             **if** *error* = NONE **and** VERIFY(*output*, *answer_gold*) **then**
20:                 *solver_code* ← *CODE*
21:                 *value_ranges* ← LLM_ANALYZE_VALUE_RANGES(*problem_text*, *numeric_var*, *solver_code*)
22:                 **break**
23:             **end if**
24:             **if** *refine_step* = *max_refine* **then**
25:                 **break**
26:             **end if**
27:             *PROMPT_refine* ← BUILDPROMPTCODEREFINE(*problem_text*, *answer_gold*, *history*)
28:             *CODE* ← LLM_CALL(*PROMPT_refine*)
29:         **end for**
30:         **if** *solver_code* ≠ FAILURE **then**
31:             **break**
32:         **end if**
33:     **end for**
34:     **if** *solver_code* = FAILURE **then**
35:         **return** NONE
36:     **end if**
37:     *new_value* ← SAMPLE(*value_ranges*)
38:     *new_problem_text* ← INSTANTIATENUMERICPROBLEMTEXT(*problem_text*, *numeric_var*, *new_value*)
39:     (*new_answer_gold*, *error*) ← RUNPYTHON(*solver_code*, input = *new_value*)
40:     **if** *error* ≠ NONE **then**
41:         **return** NONE
42:     **end if**
43:     *ok* ← LLM_FINAL_CHECK(*problem_text*, *answer_gold*, *solution_sketches*, *solver_code*, *numeric_var*, *value_ranges*, *new_value*, *new_problem_text*, *new_answer_gold*)
44:     **if** *ok* **then**
45:         **return** (*new_problem_text*, *new_answer_gold*)
46:     **end if**
47:     **return** NONE
48: **end function**

---

An example of numeric substitutions is given as follows.

- **Original:**

    Find the largest possible real part of $(75 + 117i)z + \frac{96+144i}{z}$ where $z$ is a complex number with $|z| = 4$. A common shortcut is to take $z$ to be a positive real number, since for a fixed modulus the real part is often largest when the argument of $z$ is zero.

- **Numeric variant:**

    Find the largest possible real part of $(100 + 112i)z + \frac{60+144i}{z}$ where $z$ is a complex number with $|z| = 4$. A common shortcut is to take $z$ to be a positive real number, since for a fixed modulus the real part is often largest when the argument of $z$ is zero.

**Analogical: Conditional Recomposition**

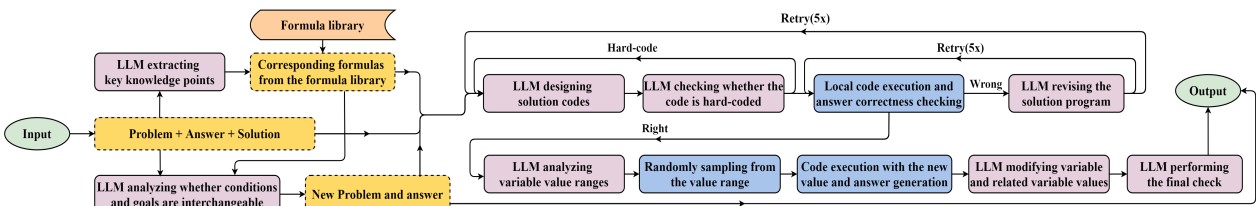

*Figure 9.* Conditional Recompositions Flow Chart.

For conditional recompositions, we again adopt a general and automatable pipeline:

- Invoke the LLM to extract the primary knowledge points tested by the problem.
- Query a pre-constructed formula library indexed by knowledge point to retrieve relevant formulas.
- Invoke the LLM to assess whether the conditions and goal are interchangeable; if so, have it generate a condition–goal swapped problem statement, along with the corresponding new answer (i.e., the original conditions).
- The new problem and answer can be output directly, or passed to the back-end of the numeric-substitution pipeline to generate additional variants.

---

**Algorithm 5** Conditional recomposition

---

1: **function** GENERATECONDITIONALRECOMPOSITIONVARIANT(*problem_text*, *answer_gold*, *solution_sketches*, *gen_variants*, *max_iter*= 5, *max_refine*= 5)
2:     *knowledge_points* ← LLM_EXTRACT_KNOWLEDGE_POINTS(*problem_text*, *solution_sketches*)
3:     *retrieved_formulas* ← RETRIEVEFORMULAS(*knowledge_points*)
4:     *history* ← EMPTY_LIST()
5:     *ANALYSIS_PROMPT* ← BUILDPROMPTINVERTIBLEANALYSIS(*problem_text*,*answer_gold*,*solution_sketches*,*retrieved_formulas*)
6:     (*invertible*, *cond_as_unknown*, *target_as_given*, *recomposed_problem_text*) ← LLM_CALL(*ANALYSIS_PROMPT*)
7:     **if** *invertible* = FALSE **then**
8:         **return** NONE
9:     **end if**
10:     **if** *gen_variants* = FALSE **then**
11:         **return** (*recomposed_problem_text*, *cond_as_unknown.value*)
12:     **end if**
13:     *solver_code* ← FAILURE
14:     *value_ranges* ← NONE
15:     **for** *iter* = 1 **to** *max_iter* **do**
16:         *PROMPT* ← BUILDPROMPTRECOMPOSEDCODEGEN(*origin_problem_text* = *problem_text*, *new_problem_text* = *recomposed_problem_text*, *origin_answer_gold* = *answer_gold*, *new_answer_gold* = *cond_as_unknown.value*, *solution_sketches* = *solution_sketches*, *retrieved_formulas* = *retrieved_formulas*)
17:         APPEND(*history*, (*PROMPT*, NONE))
18:         *CODE* ← LLM_CALL(*PROMPT*)
19:         *is_hard_code* ← LLM_HARD_CODE_CHECK(*CODE*)
20:         **if** *is_hard_code* **then**
21:             **continue**
22:         **end if**
23:         **for** *refine_step* = 0 **to** *max_refine* **do**
24:             (*output*, *error*) ← RUNPYTHON(*CODE*, input = *answer_gold*)
25:             APPEND(*history*, (*CODE*, (*output*, *error*)))
26:             **if** *error* = NONE **and** VERIFY(*output*, *cond_as_unknown.value*) **then**
27:                 *solver_code* ← *CODE*
28:                 *value_ranges* ← LLM_ANALYZE_VALUE_RANGES(*recomposed_problem_text*, *target_as_given*, *solver_code*)
29:                 **break**
30:             **end if**
31:             **if** *refine_step* = *max_refine* **then**
32:                 **break**
33:             **end if**
34:             *PROMPT_refine* ← BUILDPROMPTRECOMPOSEDCODEREFINE(*recomposed_problem_text*, *cond_as_unknown.value*, *history*)
35:             *CODE* ← LLM_CALL(*PROMPT_refine*)
36:         **end for**
37:         **if** *solver_code* ≠ FAILURE **then**
38:             **break**
39:         **end if**
40:     **end for**
41:     **if** *solver_code* = FAILURE **then**
42:         **return** NONE
43:     **end if**
44:     *new_value* ← SAMPLE(*value_ranges*)
45:     *new_problem_text* ← INSTANTIATERECOMPOSEDPROBLEMTEXT(*recomposed_problem_text*, *target_as_given*, *new_value*)
46:     (*new_answer_gold*, *error*) ← RUNPYTHON(*solver_code*, input = *new_value*)
47:     **if** *error* ≠ NONE **then**
48:         **return** NONE
49:     **end if**
50:     *ok* ← LLM_FINAL_CHECK(*problem_text*, *answer_gold*, *recomposed_problem_text*, *cond_as_unknown*, *target_as_given*, *solution_sketches*, *solver_code*, *value_ranges*, *new_value*, *new_problem_text*, *new_answer_gold*)
51:     **if** *ok* **then**
52:         **return** (*new_problem_text*, *new_answer_gold*)
53:     **end if**
54:     **return** NONE
55: **end function**

---

An example of conditional recompositions is given as follows.

- **Original:**

  Rectangles $ABCD$ and $EFGH$ are drawn such that $D, E, C, F$ are collinear. Also, $A, D, H, G$ all lie on a circle. If $BC = 16$, $AB = 107$, $FG = 17$, and $EF = 184$, what is the length of $CE$?

- **Conditional recomposition:**

  Rectangles $ABCD$ and $EFGH$ are drawn such that $D, E, C, F$ are collinear. Also, $A, D, H, G$ all lie on a circle. If $BC = 16$, $AB = 107$, $CE = 104$, and $EF = 184$, what is the length of $FG$?

**Novel: Recent-source Adaptation**

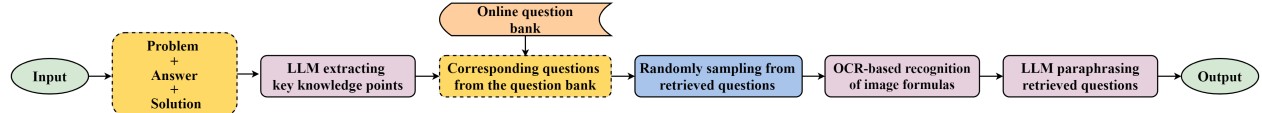

*Figure 10.* Recent-source Adaptation Flow Chart.

Recent-source generates a set of new, recent-source problems that are structurally aligned at the knowledge level but clearly distinct in surface form and provenance. The entire workflow is driven by scripts and general prompts:

- Invoke the LLM to extract the primary knowledge points tested by the problem.
- Query open online exam-question banks indexed by region, year, subject, and knowledge point, and crawl the most recent exam problems that match the extracted knowledge points.
- Invoke the LLM to paraphrase the retrieved problems, and optionally apply the three analogical transformation pipelines (redundancy insertion, numeric substitution, and conditional recomposition) to generate additional variants.

---

**Algorithm 6** Recent-Source Adaptation

---

1: **function** RECENTSOURCEADAPTATION(*problem_text*, *solution_sketches*)
2:     *KP_PROMPT* ← BUILDPROMPTEXTRACTKNOWLEDGEPOINTS(*problem_text*, *solution_sketches*)
3:     *KP_RESPONSE* ← LLM_CALL(*KP_PROMPT*)
4:     *KPs* ← PARSEKNOWLEDGEPOINTS(*KP_RESPONSE*)
5:     **if** *KPs* = NONE **then**
6:         **return** NONE
7:     **end if**
8:     *year_range* ← *RECENT_YEARS*
9:     *subject* ← *SUBJECT*
10:     *candidate_items* ← RETRIEVEEXAMS(*knowledge_points* = *KPs*, *year_range* = *year_range*, *subject* = *subject*)
11:     **if** *candidate_items* = NONE **then**
12:         **return** NONE
13:     **end if**
14:     *candidate_item* ← RANDOMSAMPLE(*candidate_items*)
15:     *processed_item* ← PROCESSCONTENT(*candidate_item*)
16:     *PARA_PROMPT* ← BUILDPROMPTPARAPHRASE(*processed_item.text*, *KPs*)
17:     *PARA_RESPONSE* ← LLM_CALL(*PARA_PROMPT*)
18:     *paraphrased_text* ← PARSEPARAPHRASEDPROBLEM(*PARA_RESPONSE*)
19:     **return** (*paraphrased_text*, *processed_item.answer_gold*)
20: **end function**

---

For example, one recent-source adaptation question generated for the concept of *logarithms* is:

Given $2^{\log_2 a} = 3$ and $\log_5 5^b = 2$, find $a - b$

**Novel: Conceptual Synthesis**

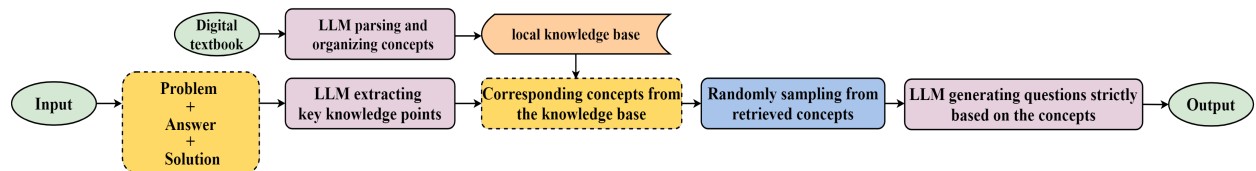

*Figure 11.* Conceptual Synthesis Flow Chart.

Conceptual synthesis turns textbook content into fresh conceptual questions that align with the original topic but are novel in form and focus. We first crawl a large collection of authoritative textbooks across different subjects from the web, and then use the LLM API's built-in functionality for parsing local PDF files to extract their content. Based on the extracted content, we build a structured knowledge base in which each concept is associated with definitions, properties, theorems, phenomena, and canonical examples extracted from the textbooks.

- Invoke the LLM to identify the primary knowledge points of the problem to be augmented, and retrieve the corresponding entries from the structured knowledge base. If the subject-specific knowledge base is missing, trigger textbook crawling and parsing to expand the knowledge base, and then retrieve the corresponding entries.
- Condition on the retrieved entries and prompt the LLM to generate new conceptual questions that target the underlying knowledge points, rather than copying any existing problem.

---

**Algorithm 7** Conceptual Synthesis

---

1: **function** CONCEPTUALSYNTHESIS(*problem_text*, *solution_sketches*)
2:     *KP_PROMPT* ← BUILDPROMPTEXTRACTKNOWLEDGEPOINTS(*problem_text*, *solution_sketches*)
3:     *KP_RESPONSE* ← LLM_CALL(*KP_PROMPT*)
4:     *KPs* ← PARSEKNOWLEDGEPOINTS(*KP_RESPONSE*)
5:     **if** *KPs* = NONE **then**
6:         **return** NONE
7:     **end if**
8:     *subject* ← *SUBJECT*
9:     *kb_entry* ← RETRIEVE_KB_ENTRY(*knowledge_points* = *KPs*, *subject* = *subject*)
10:     **if** *kb_entry* = NONE **then**
11:         CRAWL_AND_PARSE_TEXTBOOKS(*knowledge_points* = *KPs*, *subject* = *subject*)
12:         *kb_entry* ← RETRIEVE_KB_ENTRY(*knowledge_points* = *KPs*, *subject* = *subject*)
13:         **if** *kb_entry* = NONE **then**
14:             **return** NONE
15:         **end if**
16:     **end if**
17:     *GEN_PROMPT* ← BUILDPROMPTCONCEPTUALQUESTIONGENERATION(*kb_entry*, *KPs*)
18:     *GEN_RESPONSE* ← LLM_CALL(*GEN_PROMPT*)
19:     *conceptual_item* ← PARSEGENERATEDCONCEPTUALQUESTIONS(*GEN_RESPONSE*)
20:     **return** (*conceptual_item.text*, *conceptual_item.answer_gold*)
21: **end function**

---

For example, one conceptual-synthesis question generated for the concept of *logarithms* is:

What kind of mathematical idea/method turns exponentiation and multiplication into multiplication and addition?

## A.5. API Model IDs

*Table 7.* Model Name to API Model IDs Mapping.

| Model Name | API Model IDs |
| --- | --- |
| DeepSeek-V3 | deepseek-v3-250324 |
| Doubao-1.5-pro | doubao-1-5-pro-32k-250115 |
| GPT-3.5 | gpt-3.5-turbo |
| GPT-4.1 | gpt-4.1 |
| Moonshot V1 | moonshot-v1-8k |
| Kimi K2 | kimi-k2-0711-preview |
| Mistral Large 2.1 | mistral-large-2411 |
| Mistral Medium 3.1 | mistral-medium-2508 |
| Qwen-Plus | qwen-plus |
| Qwen2.5-32B | qwen2.5-32b-instruct |
| Qwen3-30B | qwen3-30b-a3b |

## A.6. Results on MMLU, GPQA

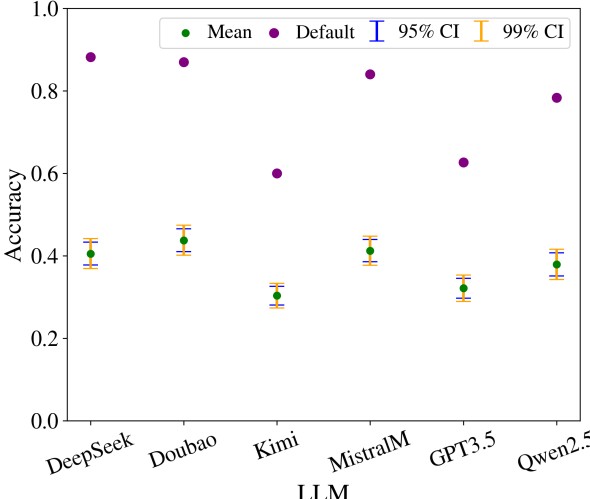

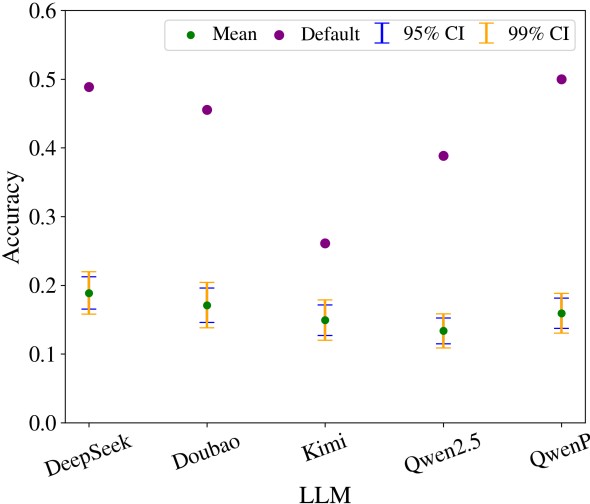

*Figure 12.* Accuracy Confidence Intervals of Different LLMs on MMLU Subset.

*Figure 13.* Accuracy Confidence Intervals of Different LLMs on GPQA Subset.

On MMLU, we observe multiple ranking reversals when comparing default-configuration point accuracies with mean accuracies estimated from systematically sampled configurations and supported by confidence intervals, e.g., DeepSeek-V3 (DeepSeek) vs. Doubao-1.5-pro (Doubao) and DeepSeek-V3 (DeepSeek) vs. Mistral Medium 3.1 (MistralM). Similar reversals arise on GPQA, including Moonshot V1 (Kimi) vs. Qwen2.5-32B (Qwen2.5), Doubao-1.5-pro (Doubao) vs. Qwen-Plus (QwenP), and DeepSeek-V3 (DeepSeek) vs. Qwen-Plus (QwenP). These results demonstrate that rankings induced by single default configurations can substantially disagree with rankings based on configuration-averaged performance with statistical uncertainty explicitly accounted for. These reversals underscore the importance of our methodology for stable and reliable model comparisons.

## A.7. Generality

A-SCS generalizes to a broad range of LLM evaluation tasks. Components such as Language, Question Paraphrase, CoT, and Multi Turn can be adjusted by modifying the prompting text across different benchmarks. Decoding parameters are API-controllable, and Question Format can be adapted or disabled. In addition, the number of options for each component and the specific option values are fully user-configurable.

For benchmarks with ground truth, A-SCS provides complete and systematic evaluations. For open-ended tasks, outcomes

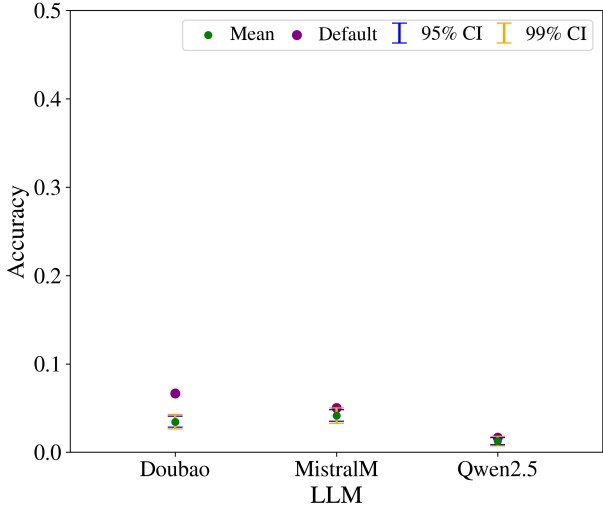

*Figure 14.* Generality Across Code Generation Benchmarks: A Case Study on A Subset of SWE-bench.

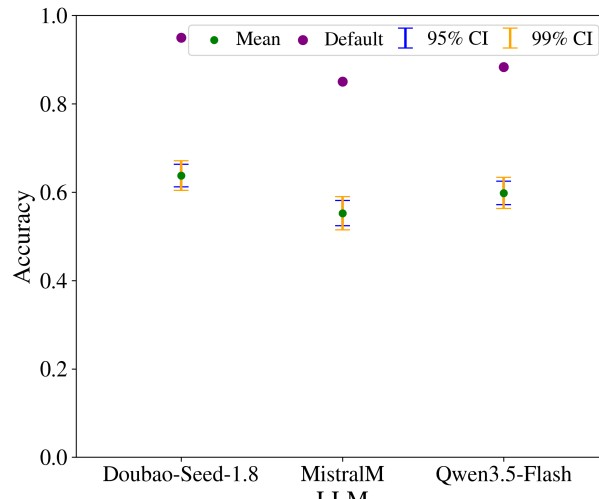

*Figure 15.* Generality Across Multimodal Benchmarks: A Case Study on A Subset of MMBench.

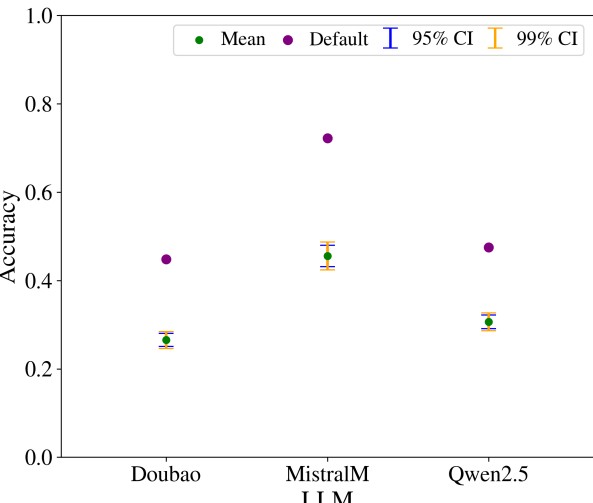

*Figure 16.* Generality Across Open-ended Benchmarks: A Case Study on A Subset of Arena-Hard.

can be derived from automatic metrics, multi-reference distributions, or model-based scoring (e.g., LLM-as-a-judge), enabling extension beyond deterministic benchmarks.

To verify the generality of our framework, we conduct additional experiments on SWE-bench (code generation), MMBench (multimodal), and Arena-Hard (open-ended evaluation), applying the same pipeline without task-specific redesign. For MMBench, we use vision-capable models, as several main-experiment models do not support multimodal input. Because full benchmark-wide evaluation is costly, we report results on randomly sampled subsets to assess whether the framework and its findings extend beyond our main text-based, closed-ended setting. Across all three benchmarks, default-configuration point estimates for most models fall outside the corresponding confidence intervals, indicating that default-setting results can be unrepresentative and potentially misleading.

## A.8. Mixed-Effects Analysis Results of A-SCS on Different LLMs

*Table 8.* Complete Mixed-effects Results on LLMs (sorted by p-value in descending order).

(a) Mixed-Effects Results on DeepSeek-V3

| Factor | $\chi^2$ | *p*-value | $\Delta \log L$ |
|---|---|---|---|
| CoT | 1388.041944 | 0.000000 | 694.020972 |
| Question Format | 370.156928 | 0.000000 | 185.078464 |
| max_tokens | 249.026022 | 0.000000 | 124.513011 |
| Language | 82.817076 | 0.000000 | 41.408538 |
| Shot | 19.223171 | 0.000012 | 9.611586 |
| top_p | 14.226885 | 0.000814 | 7.113442 |
| temperature | 13.802987 | 0.001006 | 6.901494 |
| Multi Turn | 8.824931 | 0.002971 | 4.412465 |
| presence_penalty | 6.006552 | 0.049624 | 3.003276 |
| Question Paraphrase | 0.457302 | 0.498888 | 0.228651 |

(b) Mixed-Effects Results on Doubao-1.5-pro

| Factor | $\chi^2$ | *p*-value | $\Delta \log L$ |
|---|---|---|---|
| CoT | 3218.411476 | 0.000000 | 1609.205738 |
| Question Format | 741.932770 | 0.000000 | 370.966385 |
| max_tokens | 90.865889 | 0.000000 | 45.432945 |
| Language | 49.124290 | 0.000000 | 24.562145 |
| Multi Turn | 8.108594 | 0.004406 | 4.054297 |
| Shot | 3.866155 | 0.049269 | 1.933078 |
| top_p | 5.726964 | 0.057070 | 2.863482 |
| presence_penalty | 4.988716 | 0.082549 | 2.494358 |
| Question Paraphrase | 1.035824 | 0.308795 | 0.517912 |
| temperature | 1.958708 | 0.375554 | 0.979354 |

(c) Mixed-Effects Results on GPT-3.5

| Factor | $\chi^2$ | *p*-value | $\Delta \log L$ |
|---|---|---|---|
| Question Format | 566.919850 | 0.000000 | 283.459925 |
| CoT | 383.074148 | 0.000000 | 191.537074 |
| Language | 189.867402 | 0.000000 | 94.933701 |
| top_p | 19.688407 | 0.000053 | 9.844203 |
| temperature | 8.510457 | 0.014190 | 4.255229 |
| max_tokens | 6.095081 | 0.047476 | 3.047541 |
| presence_penalty | 2.577504 | 0.275615 | 1.288752 |
| Shot | 0.421488 | 0.516196 | 0.210744 |
| Question Paraphrase | 0.240829 | 0.623608 | 0.120414 |
| Multi Turn | 0.231217 | 0.630623 | 0.115608 |

(d) Mixed-Effects Results on GPT-4.1

| Factor | $\chi^2$ | *p*-value | $\Delta \log L$ |
|---|---|---|---|
| CoT | 1367.078659 | 0.000000 | 683.539330 |
| max_tokens | 308.186114 | 0.000000 | 154.093057 |
| Question Format | 93.707267 | 0.000000 | 46.853633 |
| Language | 78.561581 | 0.000000 | 39.280790 |
| Shot | 42.520886 | 0.000000 | 21.260443 |
| Multi Turn | 26.744694 | 0.000000 | 13.372347 |
| presence_penalty | 12.264828 | 0.002171 | 6.132414 |
| top_p | 2.669075 | 0.263280 | 1.334538 |
| temperature | 2.114171 | 0.347467 | 1.057085 |
| Question Paraphrase | 0.385729 | 0.534553 | 0.192864 |

(e) Mixed-Effects Results on Qwen2.5-32B

| Factor | $\chi^2$ | *p*-value | $\Delta \log L$ |
|---|---|---|---|
| CoT | 2530.625092 | 0.000000 | 1265.312546 |
| Question Format | 789.411386 | 0.000000 | 394.705693 |
| max_tokens | 93.682979 | 0.000000 | 46.841490 |
| Shot | 62.990589 | 0.000000 | 31.495294 |
| Language | 45.532353 | 0.000000 | 22.766177 |
| Multi Turn | 18.221498 | 0.000020 | 9.110749 |
| presence_penalty | 4.736204 | 0.093658 | 2.368102 |
| top_p | 2.578881 | 0.275425 | 1.289440 |
| Question Paraphrase | 0.637877 | 0.424481 | 0.318939 |
| temperature | 0.412836 | 0.813493 | 0.206418 |

(f) Mixed-Effects Results on Qwen-Plus

| Factor | $\chi^2$ | *p*-value | $\Delta \log L$ |
|---|---|---|---|
| CoT | 2529.717264 | 0.000000 | 1264.858632 |
| Question Format | 277.842034 | 0.000000 | 138.921017 |
| max_tokens | 92.684951 | 0.000000 | 46.342475 |
| Language | 51.659881 | 0.000000 | 25.829940 |
| Shot | 34.281288 | 0.000000 | 17.140644 |
| presence_penalty | 17.094192 | 0.000194 | 8.547096 |
| temperature | 13.614385 | 0.001106 | 6.807193 |
| Multi Turn | 6.516765 | 0.010686 | 3.258382 |
| top_p | 2.607890 | 0.271459 | 1.303945 |
| Question Paraphrase | 0.182523 | 0.669214 | 0.091261 |

(g) Mixed-Effects Results on Mistral Large 2.1

| Factor | $\chi^2$ | *p*-value | $\Delta \log L$ |
|---|---|---|---|
| CoT | 1633.777359 | 0.000000 | 816.888680 |
| Question Format | 411.460934 | 0.000000 | 205.730467 |
| max_tokens | 93.434087 | 0.000000 | 46.717044 |
| Multi Turn | 25.046320 | 0.000001 | 12.523160 |
| Language | 34.192910 | 0.000002 | 17.096455 |
| Shot | 18.114376 | 0.000021 | 9.057188 |
| presence_penalty | 2.230299 | 0.327866 | 1.115149 |
| temperature | 2.133581 | 0.344111 | 1.066791 |
| top_p | 0.525114 | 0.769083 | 0.262557 |
| Question Paraphrase | 0.017725 | 0.894088 | 0.008862 |

(h) Mixed-Effects Results on Mistral Medium 3.1

| Factor | $\chi^2$ | *p*-value | $\Delta \log L$ |
|---|---|---|---|
| max_tokens | 1674.248049 | 0.000000 | 837.124025 |
| CoT | 1319.096791 | 0.000000 | 659.548396 |
| Question Format | 454.475928 | 0.000000 | 227.237964 |
| Language | 76.128528 | 0.000000 | 38.064264 |
| Shot | 19.344495 | 0.000011 | 9.672248 |
| Multi Turn | 15.965356 | 0.000065 | 7.982678 |
| Question Paraphrase | 3.119380 | 0.077366 | 1.559690 |
| presence_penalty | 4.574458 | 0.101547 | 2.287229 |
| temperature | 0.954261 | 0.620561 | 0.477131 |
| top_p | 0.147701 | 0.928810 | 0.073851 |

(i) Mixed-Effects Results on Moonshot V1

| Factor | $\chi^2$ | $p$-value | $\Delta \log L$ |
|---|---|---|---|
| CoT | 247.348130 | 0.000000 | 123.674065 |
| Question Format | 208.990733 | 0.000000 | 104.495367 |
| Language | 63.624480 | 0.000000 | 31.812240 |
| max_tokens | 24.803599 | 0.000004 | 12.401800 |
| Multi Turn | 11.392501 | 0.000737 | 5.696250 |
| Shot | 3.033164 | 0.081579 | 1.516582 |
| temperature | 4.381026 | 0.111859 | 2.190513 |
| Question Paraphrase | 1.604779 | 0.205227 | 0.802390 |
| top_p | 2.630304 | 0.268434 | 1.315152 |
| presence_penalty | 1.793134 | 0.407968 | 0.896567 |

*Table 9.* Random-intercept Standard Deviations for Questions and Augmentation Pipelines Across Models.

| Model | SD (question_id) | SD (augmentation) |
|---|---|---|
| DeepSeek-V3 | 0.46302 | 0.93016 |
| Doubao-1.5-pro | 0.47407 | 0.87181 |
| GPT-3.5 | 0.39493 | 0.88337 |
| GPT-4.1 | 0.41805 | 0.94824 |
| Qwen2.5-32B | 0.50068 | 1.02253 |
| Qwen-Plus | 0.48120 | 0.92448 |
| Mistral Large 2.1 | 0.47566 | 0.88399 |
| Mistral Medium 3.1 | 0.59870 | 0.96576 |
| Moonshot-V1 | 0.51702 | 1.02877 |

## A.9. Fixed-budget Evaluation with 500 Samples

Figure 17 reports the accuracy of each model under a non-adaptive, fixed-budget evaluation where all models are run on 500 configurations–a conservative sample-size upper bound for both SCS and A-SCS. For each model, the green dot shows the mean accuracy, with blue and orange bars denoting the corresponding 95% and 99% binomial confidence intervals; the purple dot shows the accuracy under the default configuration, and the red diamond indicates the "restricted-space ground truth" obtained from exhaustive evaluation on the most influential components. The rankings of these models under the fixed 500-sample evaluation and under the model-specific sample sizes determined by the procedure in Section 4.3 remain the same. Moreover, the ground truth still lies within the confidence intervals induced by the 500-sample evaluation, indicating that the main empirical findings remain stable under the fixed-budget analysis.

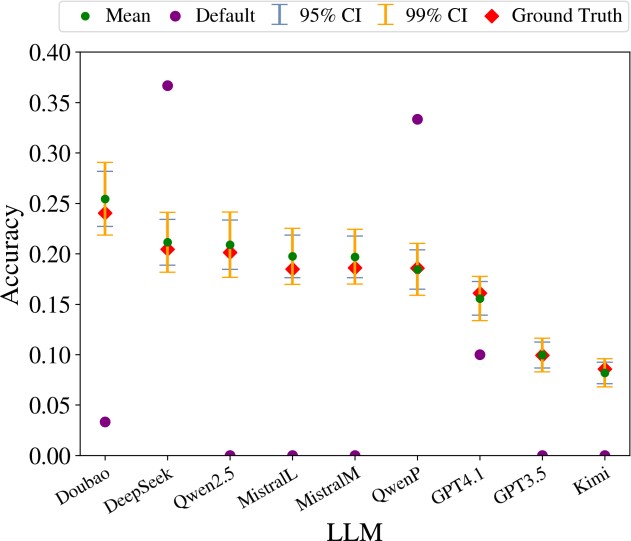

*Figure 17.* Fixed-budget Evaluation with 500 Samples per Model on AIME.

## A.10. Latency and Accuracy Analysis of LLMs

In this section, we analyze the relationships between the latency and accuracy of LLMs based on online evaluation. On one hand, we conduct online testing to assess the accuracy of LLMs under different configuration spaces in terms of the "tail to

quality" (Yang et al., 2022) metric. Here, "tail to quality" refers to the ratio of the number of questions correctly completed within a specified threshold to the total number of questions. Figure 18a illustrates the performance of various LLMs under the "Tail to Quality" metric, showing how their quality scores evolve across different threshold values. Among the models, DeepSeek-V3 (green curve) consistently demonstrates the highest quality across all thresholds, outperforming the others. Doubao-1.5-pro (blue curve) and Qwen-Plus (gray curve) follow, with Doubao-1.5-pro approaching DeepSeek-V3's performance at higher thresholds. Moonshot-V1 (brown curve) and Qwen2.5-32B (cyan curve) exhibit relatively lower quality, though Qwen2.5-32B shows rapid improvement at lower thresholds before plateauing. Overall, the chart highlights DeepSeek-V3's superior capability in handling tail data, while Qwen2.5-32B's growth in quality becomes limited at higher thresholds.

On the other hand, following a similar approach as for accuracy, we obtain the 95% and 99% confidence intervals for latency, as shown in Figure 18b. It can be seen that, for most models, latency and accuracy on AIME'2024 are positively correlated. Notably, Doubao-1.5-pro and Qwen2.5-32B achieve relatively low latency while maintaining high accuracy. In contrast, GPT-4.1 and Qwen-Plus exhibit the opposite trend: they achieve lower accuracy despite higher latency.

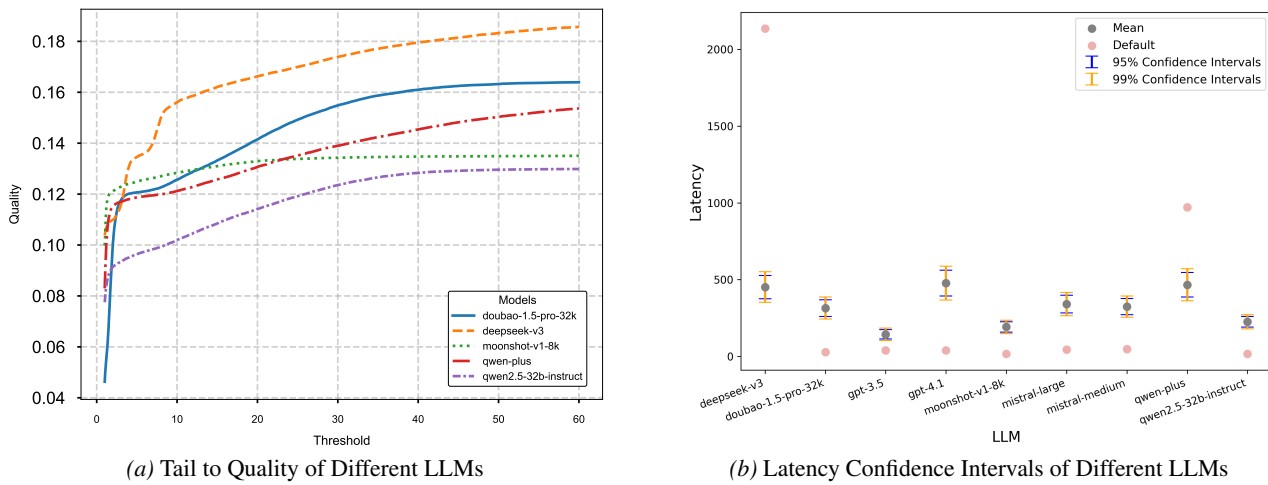

*(a)* Tail to Quality of Different LLMs         *(b)* Latency Confidence Intervals of Different LLMs

*Figure 18.* Relationship Between Inference Accuracy and Latency of LLMs in Online Testing.

