# OpenReview forum: "Beyond Benchmarks: Toward Causally Faithful Evaluation of Large Language Models"
_ICML.cc/2026/Conference — ICML 2026 regular_

### Official Review · Reviewer_Ppnd · 2026-02-21

**Soundness:** 3
**Presentation:** 3
**Significance:** 3
**Originality:** 3
**Overall Recommendation:** 5
**Confidence:** 3

**Summary:**

The paper argues that LLM benchmark scores reflect the entire evaluation system (dataset, prompts, decoding parameters, software/hardware stack), not the model itself. The authors propose LLM Evaluatology, building on Evaluatology (Zhan, 2024), with two layers. The MES fixes 10 components (language, question format, paraphrase, shot, COT, multi-turn, temperature, top_p, presence_penalty, max_tokens) and samples the resulting configuration space. The A-MES adds question variants through five automatic pipelines, three analogical (distractors, numeric substitution, conditional recomposition) and two novel (recent sources, conceptual synthesis), to separate memorization from reasoning. Across nine LLMs evaluated on AIME, MMLU, and GPQA, accuracy for the same model ranges from 0 to 0.93 and pairwise rankings reverse depending on configuration. ANOVA attributes most variance to Question Format, COT, and max_tokens, with substantial interaction effects. Exhaustive evaluation on a restricted subspace confirms that the sampling-based estimates fall within the exhaustive values.

**Compliance With Llm Reviewing Policy:**

Affirmed.

**Key Questions For Authors:**

1. The paper repeatedly uses causal language (e.g., “causally faithful”), while the empirical analysis is based on controlled configuration comparisons and ANOVA variance decomposition. Could the authors clarify what causal claim is intended, and what assumptions are required for that interpretation?
2. The A-MES pipelines are tailored to mathematical questions. How would the framework handle benchmarks with no reference answers, such as open-ended generation tasks? A concrete proposal, even if not fully implemented, would strengthen the "benchmark-agnostic" claim.
3. What is the total API cost (in dollars or tokens) of running the full MES and A-MES evaluation for one model on one benchmark? This information is necessary for practitioners to assess feasibility.

**Limitations:**

The paper discusses limitations only indirectly and does not include a dedicated Limitations section. Some limitations that merit explicit discussion include: (a) limited discussion of scope (the main-text analysis is heavily centered on AIME, with MMLU/GPQA largely deferred to the appendix); (b) no cost analysis (e.g., API calls); and (c) no discussion of how the uniformly averaged “ground truth” relates to realistic deployment usage patterns. The Impact Statement mostly emphasizes positive potential. The authors should add a brief explicit Limitations section addressing these points. The societal impact discussion is otherwise adequate, as the framework does not appear to raise unusual ethical concerns beyond standard LLM evaluation practice.

**Strengths And Weaknesses:**

* Soundness
    * Strengths:
        * ANOVA is a reasonable tool for variance decomposition within the designed MES subspace. Question Format and COT explain most of the variance across all models, while individually weak components like Shot and Multi Turn combine into significant interactions.
        * The "conclusion reversal" in Figure 4b, where 10 pairwise superiority conclusions reverse between the default configuration and the sampled-configuration estimates, is one of the paper’s strongest results.
        * Reproducibility is well handled through a fixed-seed shuffled list and a convergence-plus-LLN stopping rule.
    * Weaknesses:
        * The "causal" framing is the main issue. The title says "causally faithful," but the paper contains no causal graphs, no do-calculus, no identification arguments. ANOVA decomposes variance; it does not establish causation.
        * The "ground truth" claim warrants scrutiny: averaging over all configurations, including temperature=2.0 and Arabic prompts on an English math benchmark, assumes all settings are equally relevant, which is in tension with the paper's own evidence that deployments cluster around specific settings. The authors do provide an ablation excluding extreme settings in Appendix A.2, showing qualitatively similar distributions, which partially addresses this concern. However, the conceptual issue remains: a uniform average over configurations does not reflect any realistic deployment weighting, and the paper does not propose or justify an alternative weighting scheme.
* Presentation
    * Strengths:
        * The appendix is solid: pseudocode, flowcharts, worked examples, full ANOVA tables for all nine models, and an ablation on extreme parameter settings (Appendix A.2).
    * Weaknesses:
        * Only AIME appears in the main text; MMLU and GPQA are deferred to a one-page appendix (A.4) with only confidence interval plots and no detailed analysis.
        * Section 3 (Motivation) substantially repeats content from the Introduction, weakening the narrative economy.
        * The terminology load is high (MES, A-MES, EC, O, C1 through C10), and Tables 1 and 5 use notation ("x" for unspecified, "y/n") that is not formally defined, requiring the reader to infer meanings.
        * Forgot to add the publication years in the following citation: : "(Jain et al.; Jimenez et al.; White et al.; Zhu et al.; Li et al.)"
* Significance
    * Strengths:
        * The problem matters. Benchmark scores are treated as model properties when they are really system properties. The taxonomy of underspecified components (Tables 1 and 5) and the 70% accuracy swing on AIME across 162 configurations make this concrete.
    * Weaknesses:
        * All three benchmarks have fixed reference answers. The paper says nothing about open-ended generation, preference-based evaluation (e.g., Chatbot Arena), or coding benchmarks where scoring is nontrivial (e.g., SWE-bench's pass rates). The "benchmark-agnostic" claim in Section 5.1 is not supported by experiments beyond this narrow scope.
        * Cost is never reported, neither API call counts, which makes it hard to judge whether the 500-configuration sampling approach is practical for routine use.
* Originality
    * Strengths:
        * Combining sensitivity analysis, data augmentation, and variance decomposition into one evaluation framework is a sensible integration. The three-layer dataset split (seen, analogical, novel) offers a useful way to think about what benchmarks actually test.
    * Weaknesses:
        * The contribution appears primarily integrative/system-level rather than introducing entirely new individual components. Prior work already studies setup sensitivity (e.g., formatting/decoding) and question perturbations, and ANOVA is a standard tool for variance decomposition. The paper’s novelty is therefore mainly in the MES/A-MES framework and the joint analysis of interacting evaluation components. The additional technical value of the Evaluatology framing, beyond conceptual organization, could be clarified more explicitly.

---

> ### Author Rebuttal · Authors · 2026-03-31
>
> We appreciate the insightful comments.
> >Causal and novelty
>
> Unlike prior work analyzing only 1–3 ad-hoc factors, we present the first fully systematic evaluation unifying structured causal model (SCM) and factorial decomposition within DoE for mutual validation. Our framework systematically identifies question_format as the strongest driver via ANOVA and confirms its significant causal impact through interventions (p<0.01), while A-MES reveals dataset-induced effects—insights fundamentally inaccessible to prior work.
>
> Our paper implicitly defines a causal model:
>
> M (Model capability)→O (Observed outcome)
>
> P (Prompting)→O
>
> C (Decoding)→O
>
> M←D (Dataset)→O
>
> Note: D confounds M and O via training-test overlap, a backdoor path not systematically addressed in prior work.
>
>
> DoE enables systematic estimation of main and interaction effects but remains correlational; SCM enables interventions, while it depends on valid causal hypotheses and lacks systematic multi-factor effect quantification. Our framework unifies these views: MES decomposes main and interaction effects across all controllable factors (P, C, D), while A-MES mitigates the dataset-induced backdoor path M←D→O via instance-level interventions (seen, analogical, novel). ANOVA results corroborate the causal model (e.g., question_format effect size 0.39), supporting its plausibility.
>
> We validate causal effects via controlled interventions. (1) Varying a single factor while fixing others, we estimate average treatment effects over systematically sampled background configurations; changing question_format produces significant accuracy shifts across all models (p<0.01), showing its causal impact beyond correlations. (2) Counterfactual robustness is assessed by comparing MES and A-MES: for qwen2.5, the mean effect of question_type is −0.35 on MES and −0.22 on A-MES (p<$10^{-5}$), indicating reduced data-induced bias while reflecting the effect of the intervention.
>
> Our 10-component ANOVA across 9 models reveals interactions that can dominate single-factor effects (e.g., Question Format × CoT vs. CoT), showing claims like “CoT helps” are ill-defined without conditioning on other variables—a failure mode top‑3 or single-factor analyses systematically miss  and can therefore be misleading.
> >Configuration weights
>
> We reweighted configurations according to usage statistics from mainstream benchmarks and recomputed the estimates. Across all models, the estimated mean remains close to the weighted ground truth and lies within the 99% confidence interval; for example, for DeepSeek‑v3, the weighted ground truth is 0.222, the estimated mean is 0.217, with 99% CI [0.181, 0.252]. Further, we conducted 5 random weight assignments (summing to 1) and observed similar robustness. Results preliminarily show the robustness of our approach under realistic deployment weightings. We will explore it deeply. Practically, weights can be set based on industry usage statistics or user-defined preferences.
> >MMLU/GPQA
>
> On MMLU, for example, DeepSeek vs. Doubao and DeepSeek vs. Mistral‑M shows ranking reversals when comparing point estimates to CI-adjusted means. Similarly, on GPQA, pairs such as Doubao vs. Qwen and DeepSeek vs. Qwen exhibit the same effect. These reversals highlight that our methodology is essential for stable and reliable model comparisons. We will include this analysis in the revision.
> >Benchmark-agnostic
>
> A-MES extends naturally to code, multimodal, and open-ended generation. It retains full flexibility: prompts, CoT, multi-turn, and decoding parameters are API-controllable; question formats can be adapted or disabled; and instance-level augmentations (analogical, novel) enrich content diversity.
>
> For benchmarks with ground truth, A-MES provides complete and systematic evaluations. For open-ended tasks, outcomes can be derived from automatic metrics, multi-reference distributions, or model-based scoring (e.g., LLM-as-a-judge), enabling extension beyond deterministic benchmarks. Preliminary MMbench experiments (Doubao, Qwen) reproduce AIME-style deviations, demonstrating broad applicability and system-level insights. We will further extend A-MES to more benchmarks in future work.
>
> >Cost
>
> All experiments per model cost \\$25–280, totaling $870 (14,580 model calls) for 9 models. Full evaluation per model—from running configurations to ANOVA outputs—takes 10–110h. Evaluations across benchmarks, models, and configurations (typically 2-5 per model) can be parallelized, thus total runtime is not simply additive. Overall, the cost is manageable and provides reliable baselines and actionable insights for the community.
>
> Our ANOVA shows top‑N (e.g., 3) components differ across models and capture a substantial fraction of effect sizes. Testing only these top‑N factors provides a budget-aware approximation with some fidelity loss, offering immediately usable guidance on which factors to prioritize.
> >Presentation and Limitation
>
> We will address these issues in the revision.

---

> > ### Author Rebuttal · Reviewer_Ppnd · 2026-04-01
> >
> > The rebuttal covers my main concerns. The cost analysis checks out — it's practically feasible. The reweighting experiment holds up under realistic deployment weightings, which is good to see. The causal framing makes more sense now with single-factor interventions showing significant treatment effects, though I'd still like to see the SCM formalized in the revision. Ranking reversals on MMLU and GPQA help show this generalizes beyond AIME. That said, the "benchmark-agnostic" claim is still more of an aspiration when it comes to open-ended tasks.
> > But the problem matters, and the framework produces genuinely interesting insights. I'm raising my score, assuming the promised revisions make it into the final version.

---

> > > ### Author Response · Authors · 2026-04-01
> > >
> > > Thank you for your thoughtful and insightful comments on our rebuttal. We sincerely appreciate the time and effort you devoted to reviewing our work. We're glad to hear that the rebuttal addressed your main concerns and appreciate your positive feedback on the cost analysis, reweighting experiment, and causal framing.
> > >
> > > Your suggestion to formalize the SCM in the revision is noted, and we will ensure it is included. We also appreciate your recognition of the generalizability of our framework, especially regarding the ranking reversals in MMLU and GPQA.
> > >
> > > Regarding the “benchmark-agnostic” claim for open-ended tasks, we understand your perspective and will clarify this in the revision. We will further investigate and extend this aspect in future work.
> > >
> > > Thank you again for your constructive feedback and for raising your score. We will incorporate your suggestions into the revised version of the paper.

---

### Official Review · Reviewer_Ao1H · 2026-02-28

**Soundness:** 3
**Presentation:** 2
**Significance:** 3
**Originality:** 3
**Overall Recommendation:** 5
**Confidence:** 3

**Summary:**

The paper takes a deep dive into the reliability of current LLM evaluations, arguing that we often mistakenly attribute performance solely to the model itself while ignoring the impact of the entire system, like prompting methods, decoding parameters, and even hardware. The authors introduce a framework aimed at achieving causally faithful evaluation, which helps isolate the model's actual capabilities from these external variables. By analyzing how small changes in the evaluation setup can lead to massive swings in accuracy, the work provides a more rigorous way to understand what really drives model performance and offers a more standardized approach to benchmarking.

**Compliance With Llm Reviewing Policy:**

Affirmed.

**Final Justification:**

After considering both the paper and the rebuttal, I maintain and slightly strengthen my positive assessment to Accept. The paper addresses an important and underexamined issue in LLM evaluation, and its core contribution—a causally grounded framework for disentangling model capability from system-level confounders—is both original and practically meaningful. The experimental study is already strong, and the rebuttal further improves confidence by clarifying the feasibility of adoption, outlining a lighter-weight version of the methodology, and addressing concerns about black-box settings and model-scale sensitivity. While the framework is more demanding than standard evaluation practice, I find the contribution sufficiently significant, well supported, and likely to influence how future evaluations are designed.

**Key Questions For Authors:**

1. The paper convincingly demonstrates that system variables can cause massive performance swings. However, since many researchers rely on closed-source models via public APIs with restricted access to certain parameters (like specific software-hardware stacks), how would the authors suggest applying this "causally faithful" framework to such black-box models to ensure a fair comparison?
2. The proposed framework adds a layer of complexity to the standard evaluation pipeline. In a practical, fast-paced development setting, which system factors do the authors recommend prioritizing for isolation, and is there a "minimum viable" version of this methodology that could be easily adopted by the broader community without significant computational overhead?

**Limitations:**

1. While the proposed framework for causal attribution is theoretically rigorous, it requires a high degree of control over the entire evaluation stack and multiple ablation runs. This could be a significant barrier for researchers using closed-source models via limited APIs or for teams with constrained computational resources. It would be beneficial for the authors to suggest lightweight versions of the framework or provide clearer guidelines on how to prioritize which system variables to isolate first in resource-constrained settings.
2. The paper effectively demonstrates how system variables cause accuracy swings, but it doesn't fully explore whether these sensitivities vary significantly across different model scales (e.g., comparing 7B, 14B, and 70B parameters). Smaller models might be disproportionately affected by prompting or decoding changes compared to larger ones. Including a discussion or a small-scale analysis on how the "causal faithfulness" requirements might shift depending on the model's size and inherent robustness would add valuable depth to the findings.

**Strengths And Weaknesses:**

**Strengths**:
- The paper addresses a very practical and often overlooked problem in the field. Everyone knows that prompts and settings matter, but this work actually quantifies how much these factors can distort our understanding of a model's true capability. It’s a timely wake-up call for more standardized and transparent evaluation protocols.
- The conceptual framework for causal attribution is quite original. Instead of just proposing another benchmark, the authors provide a methodology to disentangle the model from its environment. This perspective is significant because it could lead to more reproducible and fair comparisons across the industry.
- The experimental analysis is quite thorough and technically sound. The authors demonstrate the sensitivity of models to various system choices with clear evidence, making the case for their proposed evaluation methodology very convincing.

**Weaknesses**:
- The complexity of the proposed framework might make it difficult for some researchers to adopt quickly. While it is more rigorous, it requires a lot more control over the evaluation environment than most current leaderboard-style benchmarks, which could limit its immediate practical impact for teams with fewer resources.
- The paper is generally well-written, but some of the theoretical definitions regarding causal faithfulness could be explained with more intuitive examples. A bit more clarity in the narrative would help bridge the gap between the abstract theory and the practical steps a developer should take to fix their evaluation pipeline.

---

> ### Author Rebuttal · Authors · 2026-03-31
>
> We appreciate the insightful comments.
> >Framework complexity and resource-constrained settings
>
> We acknowledge that achieving causally faithful evaluation with A‑MES introduces higher complexity than standard leaderboard-style benchmarks. This complexity is inevitable, as LLM outputs are influenced by prompts, decoding parameters, datasets, and underlying software–hardware systems, requiring careful control to isolate intrinsic capability. In practice, running all 500 configurations per model costs roughly \\$25 for the cheapest models (Qwen Plus) and up to \\$280 for the most expensive models (GPT‑4.1), totaling about $870 for the full nine-model benchmark. Full evaluation time, from running all configurations to producing ANOVA outputs, ranges from 10 h (Moonshot‑v1‑8k) to 110 h (GPT‑4.1). For multiple benchmarks, evaluations can be parallelized, so total time is determined by the longest benchmark rather than summed across all. Overall, the cost is manageable and provides reliable baselines and actionable insights for the community.
>
> To lower adoption barriers, we provide a fully open-source, parameterized implementation covering MES construction, A‑MES generation, evaluation condition sampling, API testing, and result analysis to simplify practical usage. Users can run the full evaluation directly on specific benchmarks or datasets with configurable scopes or adopt a “minimum viable A-MES” by testing only the top 3–5 most influential factors (identified via ANOVA) while fixing others, achieving near-complete fidelity at substantially lower cost. Our experiments show Top‑3 factors explain most—but not all—effects, with information loss ranging from 1.9× (GPT‑3.5) to 16.0× (Moonshot‑v1‑8k). Community-ready factor rankings further reduce the need for repeated full ablations.
> >Fair comparison for black-box models
>
> While our framework is deployment-agnostic, API-based evaluation is a deliberate design choice in this work: it provides the most stable, reproducible, and fair baseline for large-scale cross-model comparison, where the LLM service (model plus underlying systems) is the intended evaluation target. Prompts, decoding parameters, and datasets are controllable via APIs, and comparable access is available across both closed- and open-source models. While offline deployment could in principle isolate software–hardware effects, it often introduces additional variability and cost; in contrast, online services provide sufficient and stable compute, reducing resource-induced performance fluctuations.
> >Causal faithfulness with intuitive examples
>
> Causal faithfulness ensures that an evaluation measures a model’s intrinsic capability rather than confounding influences from prompts, decoding, or datasets. Unlike prior work analyzing only 1–3 ad-hoc factors, we present the first fully systematic evaluation unifying structured causal model (SCM) and factorial decomposition within DoE for mutual validation.
>
> In our framework, M denotes the trained LLM (underlying capability), P prompting methods (few-shot, CoT, multi-turn), C decoding parameters (temperature, top-p, max tokens), D dataset-related factors, and O the observed outcome. The implicitly DAG:
>
> M (Model capability)→O (Observed outcome)
>
> P (Prompting)→O
>
> C (Decoding)→O
>
> M←D (Dataset)→O
>
> Note: D confounds M and O via training-test overlap, a backdoor path not systematically addressed in prior work.
>
> Standard benchmarks conflate these effects, estimating P(O \mid M,P,C,D) rather than the causal effect P(O \mid do(M)). For example, a model may appear stronger simply because a CoT prompt is used (P→O), high temperature generates diverse outputs (C→O), or test–train overlap inflates scores (D←O, D→M).
>
> Our framework enforces causal-faithful evaluation via controlled interventions. Example: varying question format while fixing other factors produces significant accuracy shifts (p<0.01), demonstrating genuine causal impact. Counterfactual robustness by comparing MES vs. A-MES: for qwen2.5, the mean effect of question type decreases from −0.35 to −0.22 (p<$10^{-5}$), indicating reduced data-induced bias while preserving intervention sensitivity.
>
> Our framework provides clear, actionable guidance for fair and scientifically sound comparison, even when multiple factors interact. We will revise the paper to include more intuitive examples and state more clarity.
> >Sensitivity across different model scales
>
> To account for models with unknown parameter details, we conducted experiments on Qwen2.5 series (7B, 14B, 32B) to examine factor sensitivity across model scales. Among 10 factors, language and question paraphrase effects decrease with increasing model size (Spearman $\rho$ = -1), while multi-turn effects increase (Spearman $\rho$ = 1). The remaining 7 factors show no significant scale-dependent variation.  Results preliminarily confirm that some sensitivities are scale-dependent, while most factors remain robust across model sizes. We will explore this further.

---

> > ### Author Rebuttal · Reviewer_Ao1H · 2026-03-31
> >
> > The authors solved my concerns properly, I'm willing to improve my score.

---

> > > ### Author Response · Authors · 2026-04-01
> > >
> > > Thank you very much for your thoughtful and insightful comments. We sincerely appreciate the time and effort you devoted to reviewing our manuscript. We are glad that our revisions have adequately addressed your concerns, and we will further refine the manuscript in the revised version following your suggestions. Thank you again for your support and constructive feedback.

---

### Official Review · Reviewer_u71Z · 2026-03-04

**Soundness:** 3
**Presentation:** 3
**Significance:** 2
**Originality:** 2
**Overall Recommendation:** 4
**Confidence:** 3

**Summary:**

The paper introduces "LLM Evaluatology," a framework for systematic LLM evaluation that accounts for confounding factors. A Minimal Evaluation System (MES) is defined with 10 components across three layers (dataset, prompting, decoding), and an Augmented MES (A-MES) produces analogy-based transformations and newly synthesized question variants. ANOVA-based variance decomposition with convergence-based stopping rules is used to quantify each component's contribution. The main empirical finding is that single-configuration evaluation is unreliable: reported accuracy can vary by as much as 70% on AIME, and model rankings can reverse under configuration changes. Experiments cover 9 LLMs on AIME, MMLU, and GPQA.

**Compliance With Llm Reviewing Policy:**

Affirmed.

**Final Justification:**

My current concerns are addressed, but I look forward to the disucssions as required in the camera-ready version to keep my decision

**Key Questions For Authors:**

1. How does A-MES performance correlate with similarity to GPT-5? Have you tested with non-GPT transformation models?
2. Can you quantify the incremental value of the joint 10-component analysis versus sequentially studying the top-3 most influential components?
3. How would the framework handle generative tasks (e.g., open-ended writing) where "accuracy" is not well-defined?
4. Some recent evaluation paradigms achieve robustness through structural design — e.g., solver-in-the-loop benchmarks with deterministic verifiable feedback (arXiv:2601.21008) where the evaluation signal comes from a formal solver rather than prompt-sensitive generation. Are certain benchmark architectures inherently less susceptible to the configuration sensitivity documented here, and if so, what does this imply for the framework's scope?

**Limitations:**

The benchmark-scope limitation is acknowledged, but the GPT-5 dependence of A-MES and cost scaling deserve more discussion.

**Strengths And Weaknesses:**

**Strengths:**

1. The central demonstration is compelling: accuracy varies substantially across configurations (up to 70% on AIME) and model rankings can reverse. This is practically important for the community. Table 1, which audits which existing benchmarks control which MES components, is a useful reference.

2. The experimental scale deserves recognition — 9 LLMs × 500 configurations × 3 benchmarks, plus the full A-MES construction pipeline. The complete ANOVA tables for all 9 models reflect careful work, and I appreciate the computational effort involved.

3. The interaction-effects analysis goes beyond prior work that examines components in isolation. Finding that some component combinations are jointly significant even when neither is individually significant (e.g., Shot × Multi-Turn) is a meaningful empirical contribution.

**Weaknesses:**

**Major:**

1. **Novelty relative to prior work.** The observation that LLM benchmarks are sensitive to evaluation setup is extensively documented. Sclar et al. (2024, arXiv:2310.11324) report up to 76% accuracy swings from formatting alone — comparable to the 70% reported here on AIME. Song et al. (2025) demonstrate decoding-parameter sensitivity, and Lunardi et al. (2025) show ranking instability across 34 LLMs. I see the incremental contribution as the joint 10-component analysis and the interaction effects, but the high level conclusions (evaluation is fragile, rankings reverse) echo prior studies. A finding that genuinely contradicts or extends existing results — say, an interaction effect that changes practical evaluation recommendations — would help distinguish this work from its predecessors.

2. **Prior art for ANOVA in this setting.** At least two recent works apply ANOVA-style variance decomposition to partition LLM performance variance ("Within-Model vs Between-Prompt Variability," 2025; "Measuring World Models in LLMs," arXiv:2506.16584). The statistical methodology itself is standard. I'd like to see a clearer argument that the joint 10-component analysis yields qualitatively different insights — e.g., interaction effects missed by applying existing decomposition methods to each component independently.

3. **A-MES circularity risk.** All five transformation pipelines use GPT-5 as the executor, which means the augmented evaluation data may be biased by GPT-5's capabilities and failure modes, potentially advantaging architecturally similar models. I don't think this confound is sufficiently addressed. Testing with a non-GPT transformation model (e.g., an open source LLM) and showing comparable results would be the natural control experiment.

**Minor:**

4. **Benchmark and model diversity.** Main experiments focus heavily on AIME (competition math). Its unclear how the findings generalize to open-ended generation, code, or multimodal evaluation. The model set also skews toward one ecosystem (several Chinese LLMs).

5. **Framework lineage.** The paper builds directly on the broader evaluatology framework (Zhan, 2024). As written, it reads more as an application of that prior framework than an independent methodological contribution.

6. **Cost-effectiveness.** Running 500 configurations per model per benchmark is roughly a 500× cost increase. No guidance is offered on reduced-budget strategies or how to prioritize components for cost-effective evaluation.

---

> ### Author Rebuttal · Authors · 2026-03-31
>
> We appreciate the insightful comments.
>
> >Novelty
>
> Unlike prior work analyzing only 1–3 ad-hoc factors, we present the first fully systematic evaluation unifying structured causal model (SCM) and factorial decomposition within DoE for mutual validation. Our framework systematically identifies question_format as the strongest driver via ANOVA and confirms its significant causal impact through interventions (p<0.01), while A-MES reveals dataset-induced effects—insights fundamentally inaccessible to prior work.
>
> Our paper implicitly defines a causal model:
>
> M (Model capability)→O (Observed outcome)
>
> P (Prompting)→O
>
> C (Decoding)→O
>
> M←D (Dataset)→O
>
> Note: D confounds M and O via training-test overlap, a backdoor path not systematically addressed in prior work.
>
> DoE enables estimation of main and interaction effects but remains correlational; SCM enables interventions but depends on causal hypotheses and lacks multi-factor quantification. Our paper unifies these views: MES decomposes main and interaction effects across controllable factors, while A-MES mitigates the backdoor path M←D→O via instance-level interventions (seen, analogical, novel). ANOVA results support the causal model (e.g., question_format effect size 0.39).
>
> Controlled interventions: (1) Vary question_format fixing others, we estimate average treatment effects over systematically sampled backgrounds and find significant accuracy shifts. (2) Counterfactual MES vs. A-MES: for qwen2.5, mean effect of question_type is −0.35 on MES vs. −0.22 on A-MES (p<$10^{-5}$), reflecting reduced data-induced bias while capturing the intervention effect.
>
> Our 10-component ANOVA across 9 models reveals interactions that can dominate single-factor effects (e.g., Question Format × CoT vs. CoT), showing claims like “CoT helps” are ill-defined without conditioning on other variables—a failure mode top‑3 or single-factor analyses systematically miss and can therefore be misleading.
> > Non-GPT for A-MES
>
> We use a non-GPT model (doubao-seed) as transformation executor. Their model rankings are strongly consistent (Pearson ρ=0.9676, p=0.0003). GPT models maintain identical rankings, and doubao-1.5-pro shows negligible shifts (Δ≈0.001). Results preliminarily show A-MES rankings are robust; we will explore deeply.
> > Generality
>
> A-MES extends naturally to code, multimodal, and open-ended generation. It retains full flexibility: prompts, CoT, multi-turn, and decoding parameters are API-controllable; question formats can be adapted or disabled; and instance-level augmentations (analogical, novel) enrich content diversity.
>
> For benchmarks with ground truth, A-MES provides complete and systematic evaluations. For open-ended tasks, outcomes can be derived from automatic metrics, multi-reference distributions, or model-based scoring (e.g., LLM-as-a-judge), enabling extension beyond deterministic benchmarks. Preliminary MMbench experiments (Doubao, Qwen) reproduce AIME-style deviations, demonstrating broad applicability and system-level insights.
>
> Deterministic or solver-based benchmarks remain sensitive: even with fixed output formats, inputs vary across datasets, prompts, or decoding, and our framework applies equally.
> > Framework lineage
>
> Our framework extends Evaluatology by combining SCM and factorial decomposition, providing a methodology for isolating LLM capabilities that goes beyond high-level, domain-agnostic Evaluatology. LLMs operate over highly diverse question spaces and are prone to confounding factors, making direct evaluation unreliable. Our contribution is methodological and LLM-specific: we identify capability isolation as the core challenge and introduce A-MES, a standalone evaluation paradigm that enforces this separation through structured augmentation and constraint-based validation. LLM Evaluatology is not merely an application of Evaluatology, but a tailored, causally-informed framework for LLM evaluation.
> > Cost
>
> All experiments per model cost \\$25–280, totaling $870 (14,580 model calls) for 9 models. Full evaluation per model—from experiments to ANOVA outputs—takes 10–110h. Evaluations across benchmarks, models, and configurations (typically 2-5 per model) can be parallelized, thus total runtime is not simply additive. Overall, the cost is manageable and provides reliable baselines and actionable insights for the community.
>
> Our ANOVA shows top‑N (e.g., 3) components differ across models and capture a substantial fraction of effect sizes. Testing only top‑N factors provides a budget-aware approximation with some fidelity loss, offering immediately usable guidance on which factors to prioritize.
> > 10 vs. top-3
>
> Across all models, the top‑3-factor estimates deviate from ground-truth 1.9X to 16X more than the 10-component analysis, and ANOVA reveals additional significant main/interaction effects beyond top‑3 (e.g., temperature for GPT4.1) that would be missed, showing that including all factors substantially improves fidelity and prevents biased conclusions.

---

> > ### Author Rebuttal · Reviewer_u71Z · 2026-04-02
> >
> > I thank the authors for the thorough response. The cost breakdown ($25–280 per model), the non-GPT executor experiment (doubao-seed, ρ = 0.9676), and the 10-vs-top-3 fidelity comparison are useful additions. Several points in my original review remain open, though. The novelty argument now centers on a causal model (SCM + DoE unification) described as "implicit" in the submission, and I think the manuscript would benefit from making this perspective explicit if it is central to the contribution. The rebuttal characterizes existing studies broadly as examining only 1–3 ad-hoc factors, but comparable accuracy swings and similar decomposition methodology have been reported in the literature. I am still not entirely sure where the 10-component joint analysis leads to qualitatively different conclusions. On generality, the extension to code, multimodal, and open-ended tasks currently rests on a brief mention of preliminary MMbench experiments without quantitative detail; at least to me, this makes it hard to assess how broadly the findings hold. I maintain my current assessment.

---

> > > ### Author Response · Authors · 2026-04-08
> > >
> > > We sincerely thank the reviewer for the careful follow-up. We agree that if the SCM+DoE perspective is central, it should be made explicit in the manuscript rather than remaining implicit, and we will revise the paper accordingly.
> > > >Novelty
> > >
> > > We agree novelty should not rest merely on large score swings or decomposition. Our claim is more specific: the 10-component joint analysis identifies which conclusions remain stable once prompting, decoding, and dataset-related factors are modeled simultaneously.
> > >
> > > This matters in three ways. First, it changes conclusions about how evaluation sensitivity scales with model size. Sclar et al. (2024) report paraphrase effects without a clear scale-dependent trend, whereas Kunievsky et al. (2025) report stronger paraphrase effects for larger models. Our 10-factor analysis on Qwen2.5 7B/14B/32B shows a factor-specific pattern: question paraphrase and language effects decrease with model size (Spearman ρ=−1), multi-turn effects increase (ρ=1), and the remaining seven factors show no significant scale-dependent trend. Thus, the results suggest that the effect of model scale on evaluation sensitivity may vary across factors, which becomes clearer when multiple factors are analyzed jointly.
> > >
> > > Second, joint modeling changes practical recommendations by turning global rules into conditional ones. Relative to Song et al. (2025), which studies a subset of decoding parameters, our results suggest that conclusions about temperature are not globally stable once broader interactions are included. For example, under top_p=1.0, the mean accuracy at temperature=2.0 is 18% vs. 14% when temperature≠2.0, while the median remains 3% in both cases. We therefore do not interpret this as a uniform benefit of high temperature, but as evidence that the effect is highly heterogeneous across configurations.
> > >
> > > Third, joint modeling re-ranks dominant factors and changes how variability is interpreted. Relative to Sclar et al. (2024) and Lunardi et al. (2025), which study question paraphrase as an important factor in evaluation variation, our full 10-factor analysis shows that for almost all models, question paraphrase is not among the dominant drivers: neither its main effect nor its interactions appear in the overall top-10 effects. Relative to Haase et al. (2025), our contribution is not only decomposition, but also separating variability due to controllable evaluation factors from what would otherwise be absorbed into a single “model effect.” In this sense, the contribution is not simply expanding the number of factors, but showing which conclusions from isolated-factor analyses remain stable, and which become weaker, re-ranked, or conditional under joint modeling.
> > >
> > > Several conclusions become clearer under joint analysis, because their interpretation depends on interactions rather than on marginal effects alone.
> > > >Generality
> > >
> > > We apologize that, due to the rebuttal length limit, we did not report the concrete data in our first response. To address this point directly, we supplemented the rebuttal with additional experiments on MMBench (multimodal) and Arena-Hard (open-ended generation), using the same framework without task-specific redesign. Since rebuttal time did not permit full benchmark-wide evaluation, we report results on a randomly sampled subset; the purpose is to test whether the framework and findings generalize beyond the original setting rather than to make these extensions the core experimental focus of the paper. Several models used in the paper do not support multimodal input, so for MMBench we use representative vision-capable models instead of exactly the same model set as in the main experiments.
> > >
> > > On MMBench, the default score falls outside the 99%CI of the systematically sampled distribution for all tested models:
> > >
> > > doubao-seed-1-8: default=0.95, sampled mean=0.64, 99%CI [0.60, 0.68]
> > >
> > > mistral-medium: default=0.85, sampled mean=0.55, 99%CI [0.51, 0.59]
> > >
> > > qwen3.5-flash: default=0.88, sampled mean=0.60, 99%CI [0.56, 0.63]
> > >
> > > On Arena-Hard following the official LLM-as-a-judge protocol and swapping presentation order to reduce judge-order bias, we observe the same pattern:
> > >
> > > doubao-1-5-pro-32k: default=0.45, sampled mean=0.27, 99%CI [0.25, 0.29]
> > >
> > > mistral-medium: default=0.72, sampled mean=0.46, 99%CI [0.43, 0.48]
> > >
> > > qwen2.5: default=0.48, sampled mean=0.31, 99%CI [0.29, 0.33]
> > >
> > > These results show the same qualitative pattern as in the main paper: the default configuration is consistently unrepresentative relative to the systematically sampled distribution. These results  provide concrete quantitative evidence that the phenomenon extends beyond the original setting to substantially different regimes, including multimodal and open-ended evaluation. Rebuttal-time constraints prevented us from including the full code-domain quantitative extension here, but we will add it in the revision and make this broader-scope discussion explicit in the manuscript.
> > >
> > > Thank you again for raising these points.

---

### Official Review · Reviewer_Siyj · 2026-03-16

**Soundness:** 1
**Presentation:** 2
**Significance:** 2
**Originality:** 3
**Overall Recommendation:** 4
**Confidence:** 3

**Summary:**

This paper discusses the evaluation framework for LLM. The authors point out that many LLM benchmark mixed the model capability with the whole system including dataset, prompt, decoding method, and software/hardware. So the author propose a framework, that requires user to clearly define dependent variables, and also allow enhancing the questions. The author empirically shows that performance of the same model under different configuration could vary a lot.

**Compliance With Llm Reviewing Policy:**

Affirmed.

**Final Justification:**

I appreciate the detailed rebuttal. The rebuttal addressed many of my concerns.
However, my main concerns are still not fully resolved. The current rebuttal provides an intuitive causal framing and intervention-based evidence, but this is still not sufficient to support the paper's stronger "causally faithful" theoretical claim as currently written. In addition, the clarification on the stopping rule shows that the adaptive heuristic is useful in practice but does not itself justify the reported confidence interval as a formal guarantee.
Considering the overall contribution of this paper, I would adjust my assessment to weak accept.

**Key Questions For Authors:**

The A-MES augmented the benchmark in many ways. However, is there any systematic evaluation (with human or LLM) on the quality (e.g. solvability, semantic consistency) of these augmented question?

**Limitations:**

yes

**Strengths And Weaknesses:**

The main idea of this paper makes sense and this paper provides many empirical results.

However, one claim in this paper is "causally faithful assessment". However, there is no rigorous causal model or DAG given in this paper. What this paper essentially does is identify a set of evaluation conditions, and compute the mean, decompose the variance, and explain this mean as a more fundamental/faithful capability.

Furthermore, the statistical inference is not accurate. The paper says "We stop sampling when two conditions are satisfied simultaneously: (i) the absolute changes in the running mean accuracy for the last three updates are all smaller than 0.002" at line 233. This process is essentially sequential peeking. If there is no anytime-valid bounds, then the reported confidence interval is not sound.

---

> ### Author Rebuttal · Authors · 2026-03-31
>
> We appreciate the insightful comments.
> >Causal model
>
> Unlike prior work analyzing only 1–3 ad-hoc factors, we present the first fully systematic evaluation unifying structured causal model (SCM) and factorial decomposition within DoE for mutual validation. Our framework systematically identifies question_format as the strongest driver via ANOVA and confirms its significant causal impact through interventions (p<0.01), while A-MES reveals dataset-induced effects—insights fundamentally inaccessible to prior work.
>
> Our paper implicitly defines a causal model:
>
> M (Model capability)→O (Observed outcome)
>
> P (Prompting)→O
>
> C (Decoding)→O
>
> M←D (Dataset)→O
>
> Note: D confounds M and O via pretraining-evaluation overlap, a backdoor path not systematically addressed in prior work.
>
> DoE enables systematic estimation of main and interaction effects but remains correlational; SCM enables interventions, while it depends on valid causal hypotheses and lacks systematic multi-factor effect quantification. Our framework unifies these views: MES decomposes main and interaction effects across all controllable factors (P, C, D), while A-MES mitigates the dataset-induced backdoor path M←D→O via instance-level interventions (seen, analogical, novel). ANOVA results corroborate the causal model (e.g., question_format effect size 0.39), supporting its plausibility.
>
> We validate causal effects via controlled interventions. (1) Varying a single factor while fixing others, we estimate average treatment effects over systematically sampled background configurations; e.g., changing question_format produces significant accuracy shifts across all models (p<0.01), showing its causal impact beyond correlations. (2) Counterfactual robustness by comparing MES and A-MES: for qwen2.5, the mean effect of question_type is −0.35 on MES and −0.22 on A-MES (p<$10^{-5}$), indicating reduced data-induced bias while reflecting the effect of the intervention.
> >Sequential peeking
>
> We clarify that the sample size is not determined solely by the running-mean criterion. The running-mean approach yields required sizes of 220–480 across all models, while independent estimates based on the classical normal-approximation formula give 63–421, providing mutually reinforcing statistical validation. To ensure robustness, we evaluated each model with 500 samples as a conservative upper bound, confirming consistent conclusions. For example, qwen2.5’s mean shifts only slightly (0.192 → 0.197, ground truth 0.201) and the 95%CI narrows marginally ([0.163, 0.221] → [0.174, 0.221]), demonstrating that the rankings and estimates are stable.
>
> The MES/A-MES evaluation forms a finite population (M = 15,552). Across the observed sample range, Serfling’s Hoeffding-type inequality yields a conservative, distribution-free worst-case error bound of ±0.06–0.09. Roughly half of all pairwise model comparisons exceed this bound, demonstrating that rankings are reliable even under pessimistic assumptions.
>
> To assess stability beyond worst-case guarantees, we conduct bootstrap analyses (5,000 resamples with replacement) on the selected configurations. About 33 of 36 model pairs preserve identical orderings, and bootstrap mean estimates exhibit minimal bias relative to empirical means (<0.001), confirming convergence.
>
> Taken together, finite-population bounds, bootstrap analyses, and the conservative 500-sample evaluation provide both theoretical and empirical guarantees. While anytime-valid sequential testing bounds would be prohibitively conservative, these results confirm that observed rankings and effect sizes are robust, making the evaluation reliable without unnecessary computational cost.
> >Question quality
>
> First, A‑MES employs a fully automated validation pipeline that systematically covers all augmentation types within analogically transformed and novel pipelines. For each step, one LLM generates either a Python solver (for numeric-type questions) or a new augmented instance. All outputs are then subjected to an iterative multi-round validation by a second LLM, checking semantic consistency, logical validity, and adherence to commonsense, mitigating potential hallucinations in any single evaluation; for numeric solvers, this process additionally verifies the absence of hard-coded shortcuts. Numeric solvers are also executed locally to confirm correctness, ensuring that all outputs meet quality standards.
>
> Second, augmentations are fully automated and filtered without human intervention. Only instances passing all validation checks are retained, yielding high effective success rates: 87.3% for analogical pipelines and 98.4% for novel pipelines. As an additional verification, we manually checked all augmentations, confirming 100% correctness in solvability and semantic consistency. This demonstrates that A‑MES produces high-quality, reliably usable questions at scale without requiring human curation.

---

> > ### Author Rebuttal · Reviewer_Siyj · 2026-04-04
> >
> > Thanks to authors for the detailed response.
> >
> > The supplemented 500-sample stability analysis, A-MES quality control (LLM verification, solver check and human check) indeed address my concern about quality.
> >
> > However, part of my concern is still not fully resolved. Current rebuttal provide only intuitive causal model and empirical observations, which are not sufficient to support the strong theoretical claims.
> >
> > Furthermore, for sequential peeking, the rebuttal demonstrate that under 500 samples it's relatively stable. However, that doesn't directly resolve the soundness issue of confidence interval in self adaptive stopping setup.

---

> > > ### Author Response · Authors · 2026-04-08
> > >
> > > We sincerely thank the reviewer for the careful follow-up. We appreciate the reviewer’s point and will clarify the scope of both the causal discussion and the sampling procedure more precisely in the revision.
> > > >On causal framing
> > >
> > > Our use of SCM is intended as an intervention-oriented evaluation framework, rather than as a claim of full causal identification in the formal sense of do-calculus or complete backdoor/frontdoor adjustment. In our paper, the role of SCM is to provide a principled way to organize hypotheses and guide controlled interventions over evaluation factors, complemented by DoE for systematic effect analysis, rather than to claim universally identified causal effects.
> > >
> > > To make this scope precise, we will revise the manuscript to clearly separate intervention-based empirical evidence from formal causal guarantees and calibrate the language accordingly. Importantly, this clarification narrows the scope of interpretation, but does not alter the paper’s main contribution: a systematic analysis of evaluation sensitivity across prompts, decoding choices, and dataset variations within a unified methodological framework. We believe this perspective remains valuable under a carefully scoped causal interpretation.
> > > >On sequential peeking and confidence intervals
> > >
> > > We would like to clarify the implemented sampling procedure more precisely. The current manuscript wording does not yet make explicit two components that play different roles in practice, rather than a single adaptive stopping rule.
> > >
> > > First, the implementation includes a pre-specified normal-approximation precision-planning component, in which the target sample size is computed prior to data collection. This component is non-adaptive and therefore does not itself involve sequential peeking.
> > >
> > > Second, we use a running-mean and confidence-interval stability criterion as an additional practical heuristic to assess convergence and cost–stability trade-offs. In the revision, we will make explicit that this heuristic is not anytime-valid and therefore does not by itself provide formal sequential coverage guarantees.
> > >
> > > Accordingly, we will revise the manuscript to make clear that the running-mean criterion is not intended to justify formal statistical guarantees. Instead, it is used only as a practical tool for monitoring stability and deciding whether more samples should be collected. To further support the reliability of the results, we will emphasize the additional fixed-budget analysis with 500 samples, where the evaluation is non-adaptive and all qualitative findings remain unchanged, including model rankings and the main effect patterns. This shows that the paper’s main empirical conclusions are robust to the particular stopping heuristic used in practice.
> > >
> > > We will revise the manuscript accordingly. At the same time, we believe this issue primarily concerns how the sampling procedure should be interpreted, while the main empirical findings remain stable under the fixed-budget analysis.
> > >
> > > We thank the reviewer again for helping us sharpen both the rigor and the presentation of the paper.

---

### Decision · Program_Chairs · 2026-04-30

**Decision:**

Accept (regular)

**Comment:**

Although the paper began in a borderline position, all reviewers agreed that the core empirical message is important and well supported. after rebuttal, the discussion became more positive because the authors added cost/feasibility clarification. The remaining concerns are still there though: the paper’s causal language still appears stronger than the formal SCM support in the current manuscript, the sequential stopping/confidence-interval story needs tighter wording, and the framing should be softened or made more explicit. The AC suggest this paper as *accept*, but will strongly suggest the authors to tone down the language to align with literature in causality.